# *BRCA* mutational status shapes the stromal microenvironment of pancreatic cancer linking clusterin expression in cancer associated fibroblasts with *HSF1* signaling

Tumors initiate by mutations in cancer cells, and progress through interactions of the cancer cells with non-malignant cells of the tumor microenvironment. Major players in the tumor microenvironment are cancer-associated fibroblasts (CAFs), which support tumor malignancy, and comprise up to 90% of the tumor mass in pancreatic cancer. CAFs are transcriptionally rewired by cancer cells. Whether this rewiring is differentially affected by different mutations in cancer cells is largely unknown. Here we address this question by dissecting the stromal landscape of *BRCA*-mutated and *BRCA* Wild-type pancreatic ductal adenocarcinoma. We comprehensively analyze pancreatic cancer samples from 42 patients, revealing different CAF subtype compositions in germline *BRCA*-mutated *vs. BRCA* Wild-type tumors. In particular, we detect an increase in a subset of immune-regulatory clusterin-positive CAFs in *BRCA*-mutated tumors. Using cancer organoids and mouse models we show that this process is mediated through activation of heat-shock factor 1, the transcriptional regulator of *clusterin*. Our findings unravel a dimension of stromal heterogeneity influenced by germline mutations in cancer cells, with direct implications for clinical research.

Pancreatic ductal adenocarcinoma (PDAC) is one of the most aggressive cancer types, with a 10% 5-year survival rate[1]. Major contributors to this aggressiveness are cancer-associated fibroblasts (CAFs)[2,3]. CAFs comprise up to 90% of the cellular tumor microenvironment (TME) in PDAC, and promote tumorigenesis by elevating proliferation, invasion, and chemoresistance of cancer cells, and by remodeling the extracellular matrix (ECM)[4–6]. CAFs are functionally and phenotypically heterogeneous, and are composed of multiple subpopulations[7–11]. In PDAC, a series of studies identified three major CAF subtypes with distinct functions—a myofibroblastic subtype that expresses α-smooth-muscle-actin (αSMA; termed myCAF), an inflammatory subtype that expresses interleukin 6 (IL-6) and leukemia inhibitory factor (LIF; termed iCAF), and an antigen-presenting subtype that expresses MHC class II (apCAF)[12–15]. Another study described four CAF subtypes

with distinct functional features and prognostic impact[9], and single-cell analysis of human PDAC identified eight fibroblast clusters[16]. Moreover, cancer-associated mesenchymal stem cells were shown to secrete granulocyte-macrophage colony-stimulating factor (GM-CSF), acting as CAFs to support PDAC tumor progression[17], while CAFs of pancreatic stellate cells (PSC)-origin were demonstrated to regulate specific ECM features and to contribute to tumor stiffness[18]. Most recently, single-cell analysis of human PDAC identified a subset of LRRC15+ CAFs and showed a correlation between elevated levels of this subset and poor response to anti-PD-L1 therapy[19]. These studies and others[20–23] exposed additional complexity and diversity leading to the segregation of the three main subtypes of CAFs into multiple subpopulations[11]. Inflammatory CAFs, for example, were segregated into subpopulations based on expression of distinct cytokines and

✉e-mail: ruth.shouval@weizmann.ac.il

immune-modulatory genes, in addition to antigen-presenting modules[20,22]. These studies also highlighted the dynamic nature of CAFs, and their ability to shift between phenotypes depending on external signals[7], which could explain previous contradictory findings of both anti- and pro-tumorigenic effects of CAF depletion in PDAC[24–26].

CAFs are genomically stable, and rarely have copy number alterations or somatic mutations leading to loss of heterozygosity[27]. Yet, they are transcriptionally heterogeneous[8,13]. This heterogeneity is driven by different external cues received from neighboring cells and local environmental conditions[28,29]. For example, hypoxia was shown to induce a pro-glycolytic transcriptional program in CAFs[30], and a metabolic switch from oxidative phosphorylation to glycolysis was also shown in response to TGFβ and PDGF in an IDH3α-mediated mechanism[31]. The stress-induced transcriptional regulator Heat Shock Factor 1 (HSF1) was shown, by us and others, to play a key role in shaping CAF transcription in diverse human carcinomas, including breast, lung, gastric, and colon cancer[32–36]. HSF1 orchestrates a transcriptional program in fibroblasts that enables their reprogramming into CAFs and promotes malignancy by TGFβ and SDF1, YAP/TAZ signaling, and exosome-mediated secretion of THBS2 and INHBA[32–36]. CAF heterogeneity was also proposed to stem from different cells of origin giving rise to CAFs, including tissue-resident fibroblasts, mesenchymal stromal cells, pericytes, and adipocytes[8,17,37–41]. For example, bone-marrow-derived CAFs in breast cancer were shown to express high levels of Clusterin (Clu), and exhibit a distinct transcriptional profile compared to tissue-resident CAFs[40]. However, it is not known whether different germline mutations in the cancer cells lead to differential rewiring of CAFs and contribute to CAF heterogeneity.

In PDAC, a subset of up to 7% of the general population, and up to 20% in certain subgroups (such as patients of Ashkenazi Jewish descent), have germline mutations in the breast cancer-1 (BRCA1) and BRCA2 genes[42,43], which are part of the DNA damage homologous repair mechanism. BRCA mutations are the most prominent germline mutations associated with increased risk of developing pancreatic cancer[44]. Patients carrying these mutations, both in PDAC and in other BRCA-associated cancers (e.g. breast cancer), exhibit a higher response rate to platinum-based chemotherapy regimens and PARP inhibitors, resulting in longer than expected overall survival[43,45]. Several cell-autonomous mechanisms by which PARP inhibitors affect BRCA-mutant (BRCA-mut) cancer cells were suggested[46–48], however, additional non-cell-autonomous factors mediating the efficacy of these treatments may be pivotal in PDAC. Recent studies described distinct immune microenvironments in BRCA-mut breast, ovarian, and prostate cancers, characterized by increased infiltration of T cells and macrophages[49–52]. Since cells of the TME are considered to be genomically stable[27], this rewiring is thought to be orchestrated through non-cell-autonomous effects driven by BRCA mutations in the cancer cells. Yet the transcriptional landscape of the fibroblastic microenvironment of BRCA-mut PDAC remains uncharted. In breast cancer, we have recently identified two major CAF subtypes expressing either the marker S100A4 (also known as FSP1) or podoplanin (PDPN)[8]. The ratio of these two CAF subtypes was correlated with BRCA1/2 mutational status and with disease outcome in BRCA-mut breast cancer patients.

Given that CAFs are reprogrammed by the adjacent cancer cells, we hypothesize that different driver mutations will yield different stromal landscapes. Here, we set out to test this hypothesis in a comprehensive cohort of 42 BRCA-mut and BRCA-WT pancreatic cancer patients. Using three CAF markers—Clusterin (CLU), αSMA, and MHC class II—we identify three mutually exclusive CAF subtypes in primary pancreatic tumor resection specimens, and show that the ratio between these CAF subtypes is altered in BRCA-mut tumors compared to BRCA-WT tumors. We apply laser capture microdissection (LCM) followed by RNA sequencing to define stromal transcriptional signatures unique to BRCA-mut vs. BRCA-WT tumors. We characterize BRCA-associated stromal signatures by multiplexed immunofluorescence (MxIF) and second harmonic generation signaling (SHG). We find distinct stress response activation patterns in BRCA-mut vs. BRCA-WT tumors. In particular, we find that HSF1 is upregulated in BRCA-mut tumors. Using cancer organoids, co-cultures, and in-vivo models we show that loss of BRCA function in cancer cells leads to a transcriptional shift of PSCs from myofibroblastic to immune-regulatory Clu+ CAFs in an HSF1-dependent manner. Our findings portray distinct stromal compositions in BRCA-mut and BRCA-WT PDAC tumors with far-reaching clinical implications for early detection and for PDAC therapy.

## Results

### BRCA-mut and BRCA-WT tumors exhibit distinct CAF compositions

To dissect the stroma of BRCA-mut PDAC in comparison with that of BRCA-WT PDAC, we assembled a clinical cohort of 42 patients (27 BRCA-WT and 15 germline BRCA-mut; see Supplementary Data 1). Formalin-fixed primary tumor resection tissue, and deeply annotated demographic, clinical and pathologic data were collected for all patients in the study. In addition, genomic (MSK-IMPACT™) data and fresh-frozen tumor tissue was collected for a subset of patients. PDAC CAFs are comprised of distinct subtypes, marked by the expression of distinct proteins[9,12,13]. To test whether CAF compositions are affected by the germline mutational status of the cancer cells we assessed the distribution of several CAF markers in primary tumor resections from BRCA-WT and BRCA-mut PDAC patients. In particular, we stained for αSMA, podoplanin (PDPN), platelet-derived growth factor receptor alpha (PDGFRa), human leukocyte antigen DR isotype (HLA-DR; an MHC class II molecule), and S100A4, all of which were previously described as CAF markers in different cancer types[8,33,39,40]. We also stained for CLU, which was previously suggested as a marker of bone marrow-derived fibroblasts in breast cancer[4] (Fig. 1a, b and Supplementary Figure 1a, b). Of these proteins, three marked discrete CAF subtypes (negative for CD45 and cytokeratin), and together covered most of the stromal cells – αSMA, CLU and HLA-DR (MHC-II; Fig. 1a, b and Supplementary Figure 1c–e). S100A4 marked mostly CD45+ immune cells in this patient cohort, PDPN marked a subset of αSMA+ CAFs, and PDGFRa partially overlapped with other markers; therefore, these were not chosen for further analysis (Supplementary Figure 1a, b). αSMA is a well-known myofibroblastic marker in various carcinomas, including PDAC[53]. MHC-II was recently suggested as a marker of apCAFs in both breast and pancreatic cancers[8,12,15]. Clu was shown to be expressed by αSMA^low CAFs in breast and pancreatic cancer[8,13,40], however, the identity of these αSMA^low CAFs was not fully elucidated— in mouse models of PDAC Clu was shown to be expressed by apCAFs, whereas in human patient samples it is expressed by inflammatory CAFs[12]. Immunohistochemical analysis (IHC; Fig. 1a) and MxIF (Fig. 1b) staining demonstrated a segregation of αSMA+, CLU+ and HLA-DR+ CAFs in PDAC. Automated image analysis quantifying the relative abundance of these proteins in stromal cells (CD45-Cytokeratin-) in a subcohort of 10 BRCA-mut and 15 BRCA-WT tumors confirmed that the three CAF markers clearly mark discrete CAF subtypes (Supplementary Figure 1c–e), as shown by the low co-expression of each CAF marker with the other markers.

Next, we asked whether the global composition of immune cells, cancer cells and CAFs is different between BRCA-mut and BRCA-WT tumors. We quantified the number of CD45+ (immune) cells by MxIF and found no differences between the different genotypes (Supplementary Figure 1f). Then, we used an artificial intelligence image analysis algorithm to classify different cell populations in H&E−stained FFPE sections from patients. We found no significant differences in the percentage of CAF-rich, cancer-rich, and immune-rich regions between BRCA-mut and BRCA-WT tumors (Supplementary Figure 1g–j).

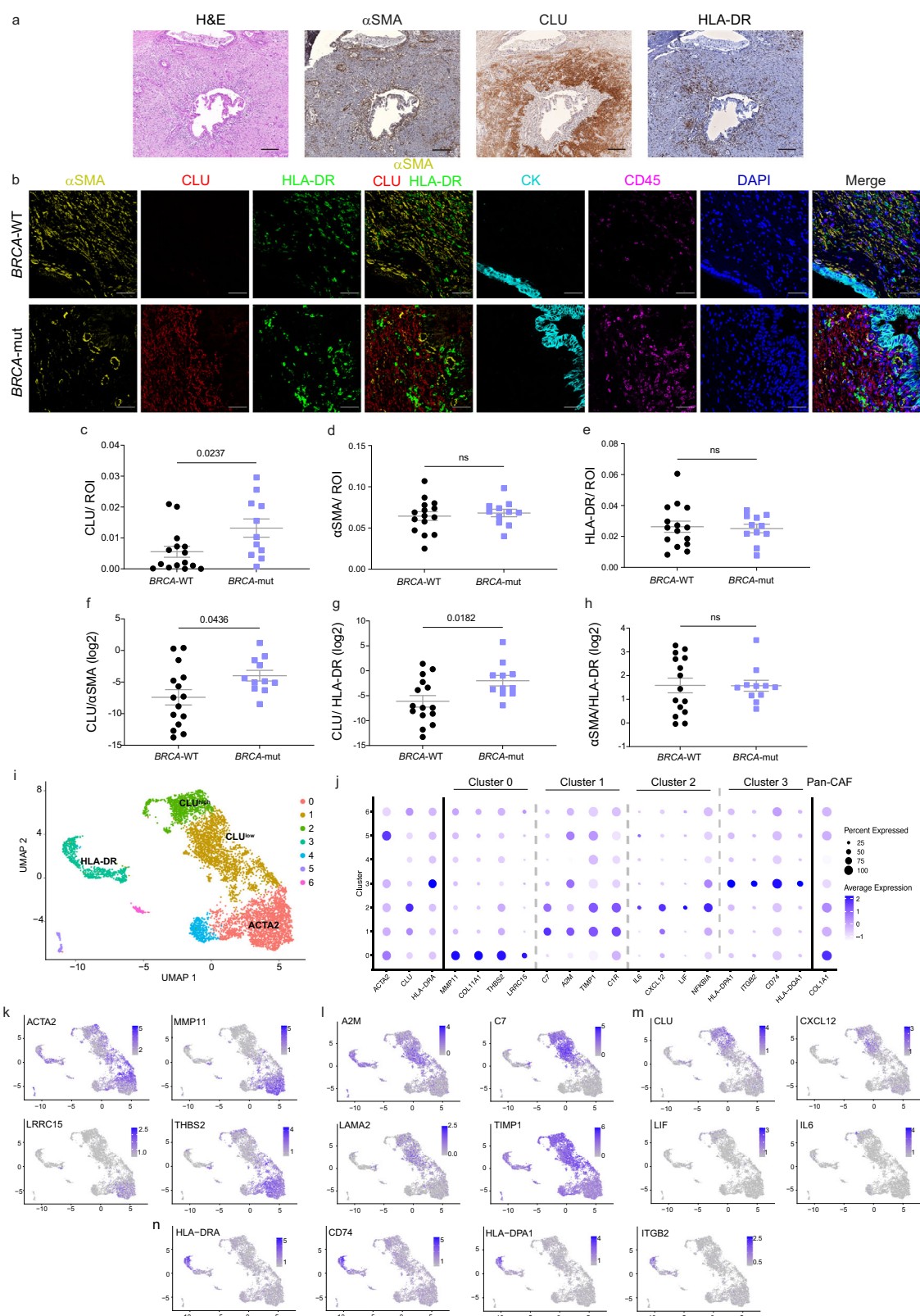

We then evaluated each subpopulation of CAFs separately, by quantifying CLU⁺, MHC-II⁺, and αSMA⁺ CAF staining in a subcohort of 26 human PDAC patients, including 15 *BRCA*-WT and 11 *BRCA*-mut patients. αSMA⁺ CAFs and HLA-DR⁺ CAFs did not differ between the genotypes. CLU⁺ CAFs were significantly more abundant in *BRCA*-mut tumors (Fig. 1c–e). Moreover, the ratio between CAF subtypes was different between *BRCA*-mut and *BRCA*-WT tumors (Fig. 1f–h).

Specifically, the ratio of CLU⁺/αSMA⁺ CAFs and the ratio of CLU⁺/HLA-DR⁺ CAFs was higher in *BRCA*-mut tumors compared to *BRCA*-WT tumors (Fig. 1f, g), suggesting that germline mutations in the cancer cells alter tumor CAF compositions.

To examine whether this characteristic is different between *BRCA1* and *BRCA2* mutant tumors, we compared the relative abundance of CLU⁺ CAFs and the CLU⁺/αSMA⁺ CAF ratio in *BRCA1 vs. BRCA2* mutant

**Fig. 1 | CAF compositions change between *BRCA*-WT and *BRCA*-mut PDAC tumors.** Formalin-fixed paraffin-embedded (FFPE) tumor sections from *BRCA*-mut and *BRCA*-WT PDAC patients were stained for hematoxylin and eosin (H&E), IHC, and MxIF. (**a**) IHC was performed for αSMA, CLU and HLA-DR (Scale bar, 200 μm). Representative images of a BRCA-WT tumor are shown (*n* = 2). (**b–h**) MxIF was performed using antibodies for the depicted proteins. DAPI was used to stain nuclei. Scale bar, 50 μm. Representative images are shown in (**b**). Images were analyzed using ImageJ software, CD45⁻ CK⁻ regions were defined as regions of interest (ROIs) and the area stained by each CAF marker was calculated, divided by the ROI and averaged for each patient sample (**c**–**e**). Mann-Whitney test was performed. The ratio of the different CAF subtypes is shown in (**f**–**h**) and was analyzed using Student's t-test. Data are presented as Mean ± SEM. ns marks p-values greater than 0.05. For IHC and H&E staining *n* = 2, and for MxIF staining *n* = 11 *BRCA*-mut and *n* = 15 *BRCA*-WT. (**i**–**n**) Single-cell RNA-seq data of fibroblasts and stellate cells from human PDAC tumors[16] was reanalyzed using the Seurat R toolkit. (**i**) Uniform Manifold Approximation and Projection (UMAP) of 6,405 cells from[16], color-coded for the indicated cell clusters defined by a local moving clustering algorithm. The clusters that differentially express *ACTA2, CLU* and *HLA-DR* are indicated. (**j**) Dot plot visualization of gene expression of the indicated CAF markers. (**k–n**) Single-cell expression level of CAF markers on the UMAP shown in (**i**). Marker genes of *ACTA2* (**k**), *CLU^low^* (**l**), *CLU^high^* (**m**), and *HLA-DR* (**n**) clusters are represented. Source data are provided as a Source Data file.

patients. We found no significant differences (Supplementary Figure 1k–l), implying that the changes in CAF compositions are shared between different *BRCA* mutations.

Several recent studies associated CLU expression with neoadjuvant therapy. One study reported elevated expression of CLU following neoadjuvant therapy in prostate cancer[54], and another suggested that low stromal expression of CLU is predictive of better response to neoadjuvant therapy in triple-negative breast cancer[55]. We therefore compared CLU⁺/αSMA⁺ CAF ratios in neoadjuvant-treated *vs.* non-treated patients. We found that the ratio of CLU⁺/αSMA⁺ CAFs was not affected by treatment (Supplementary Figure 1m). Both treated and non-treated patients had higher CLU⁺/αSMA⁺ CAF ratios in *BRCA*-mut patients compared to WT, supporting the notion that CAF distribution is driven by the tumor genotype and is not altered by neoadjuvant treatment regimens (See Supplementary Data 1 for clinical information).

Next, we sought to explore whether the CAF subtypes we characterized using protein markers could also be identified at the transcriptional level. To that end we reanalyzed data from a large and comprehensive single-cell RNA-seq dataset of human PDAC (Peng et al.[16]) using the Seurat R toolkit[56]. We reanalyzed only tumor samples (excluding cells from normal controls), and within these samples analyzed all the cells that were defined as "fibroblasts" or "stellate cells" in the original dataset, and excluded *MCAM* positive cells (a pericyte marker). Unbiased clustering of 6405 cells that passed QC (see Methods) revealed 7 distinct CAF subtypes (Fig. 1i, j, Supplementary Data 2). *ACTA2*, *CLU*, and *HLA-DR* were differentially expressed (DE) in distinct clusters, supporting our MxIF analysis and suggesting that not only at the protein level, but also at the transcriptional level, these genes mark discrete CAF populations. This segregation was evident across patients, and did not stem from intra-patient variability (Supplementary Fig. 1n). To further validate these findings, we reanalyzed an additional published single-cell RNA-seq dataset of human PDAC[12]. Here we analyzed all cells defined as iCAFs or myCAFs (HLA-DR⁺ antigen-presenting CAFs were not found in this patient dataset). Similar to the Peng dataset, in this dataset, *CLU* and *ACTA2* were differentially expressed in distinct clusters (Supplementary Figure 1o–p, Supplementary Data 3).

To further study the transcriptional signatures of these clusters we performed pathway analysis of the top DE genes in clusters that differentially expressed *ACTA2*, *CLU*, and *HLA-DR* in the Peng dataset (Fig. 1j–n; Supplementary Data 2). The *ACTA2*⁺ (αSMA; Fig. 1k, cluster 0) cluster was enriched for myofibroblastic pathways such as ECM remodeling (collagens and MMPs), wound healing (*INHBA, THBS2*), smooth muscle contraction (*ACTA2 and TPM* genes), and cell-substrate adhesion (*LRRC15, ITGB5*)[12,13,19,34]. *CLU* was differentially upregulated in two clusters—cluster 1 and cluster 2—albeit at different levels. We defined these clusters as *CLU^low^* (1) and *CLU^high^* (2), to reflect these differences. The *CLU^low^* cluster (cluster 1; Fig. 1l) was enriched with genes involved in complement and coagulation cascades (*A2M, C1R, C1S, C7*), in addition to genes involved in ECM organization (*DCN, LAMA2, TIMP1*). The *CLU^high^* cluster (cluster 2; Fig. 1m) expressed inflammatory genes (*IL-6, CXCL12, CXCL1, NFKBIA*), as well as genes

involved in ECM remodeling (*LIF, COL14A1, HAS1*) and angiogenesis regulation (*C3, IL6*)[12–14]. The *HLA-DR* cluster (cluster 3; Fig. 1n) was enriched for antigen presentation (variety of *HLA* genes), and for T cell activation (*ITGB2, S100A8*). These results indicate that CLU is a marker of a distinct CAF subset in PDAC, characterized by an immune-regulatory and inflammation-associated gene signature.

### *BRCA*-WT and *BRCA*-mut stroma exhibit distinct transcriptional signatures

To directly map the transcriptional landscapes of *BRCA*-WT and *BRCA*-mut stroma, we employed laser capture microdissection (LCM) followed by RNA-sequencing on CAF-rich regions from 12 patients (5 *BRCA*-mut and 7 *BRCA*-WT; Supplementary Data 4). Unsupervised differential expression analysis showed clear segregation between *BRCA*-WT and *BRCA*-mut tumors. This analysis revealed 30 upregulated and 10 down-regulated genes in *BRCA*-mut vs. *BRCA*-WT patients (Fig. 2a). *CLU* was not among the DE genes, however, it did show a trend of elevation in *BRCA*-mut patients compared to *BRCA*-WT patients (Supplementary Figure 2a). Pathway analysis of the differentially upregulated genes showed enrichment of genes involved in ECM remodeling and proteolysis (*MUC5B, SERPINA1, A2ML1, S100A2, GREM1*), wound healing (*TNC, CD177, WFDC1*), muscle contraction (*DES, KCNMA1, OXTR, CEMIP*), and regulation of cell growth (*CRABP2, ROS1, WFDC1*) in *BRCA*-mut *vs.* *BRCA*-WT patients (Fig. 2a and Supplementary Data 4). Genes involved in T-cell activation and migration (*IRF4, TBX21, CXCL9*) and tyrosine kinase signaling (*STAP1, FLT3*) were downregulated in *BRCA*-mut *vs.* *BRCA*-WT patients (Fig. 2a and Supplementary Data 4). To exclude the possibility that the observed differential expression of immune-related genes is due to higher immune-cell contamination of the dissected CAF-rich regions in *BRCA*-WT stroma, we applied CIBERSORTx, a computational deconvolution tool that estimates the relative abundance of individual cell types in a mixed cell population based on single-cell RNA-seq profiles[57]. First, we estimated the relative abundance of fibroblasts in our samples using the single-cell human PDAC dataset by Peng et al.[16]. We found that CAFs were predominant in all our samples, comprising 74–91% of the cells in each sample, with an average of 85%. This analysis also excluded potential cancer cell contamination and showed that there were no differences between the relative abundance of the tested cell types in *BRCA*-mut *vs BRCA*-WT samples (Supplementary Figure 2b and Supplementary Data 5). Then, we applied this tool to estimate the distribution of immune cell subtypes within these samples. Similarly, no significant differences were found between *BRCA*-WT and *BRCA*-mut stroma in any of the immune cell subtypes tested (Supplementary Figure 2c and Supplementary Data 5). Lastly, we stained a cohort of 7 *BRCA*-mut and 11 *BRCA*-WT tumors by MxIF to assess CD3 expression at the protein level and found no differences in its abundance (Supplementary Figure 2d). This converging evidence suggests that even if immune cells have infiltrated into the dissected stromal regions, the observed differential expression patterns most likely originate from CAFs.

We next set to analyze some of the DE genes at the protein level. We chose to focus on *MUC5B* and *SERPINA1*, two of the most significantly upregulated genes in *BRCA*-mut CAFs. These genes encode

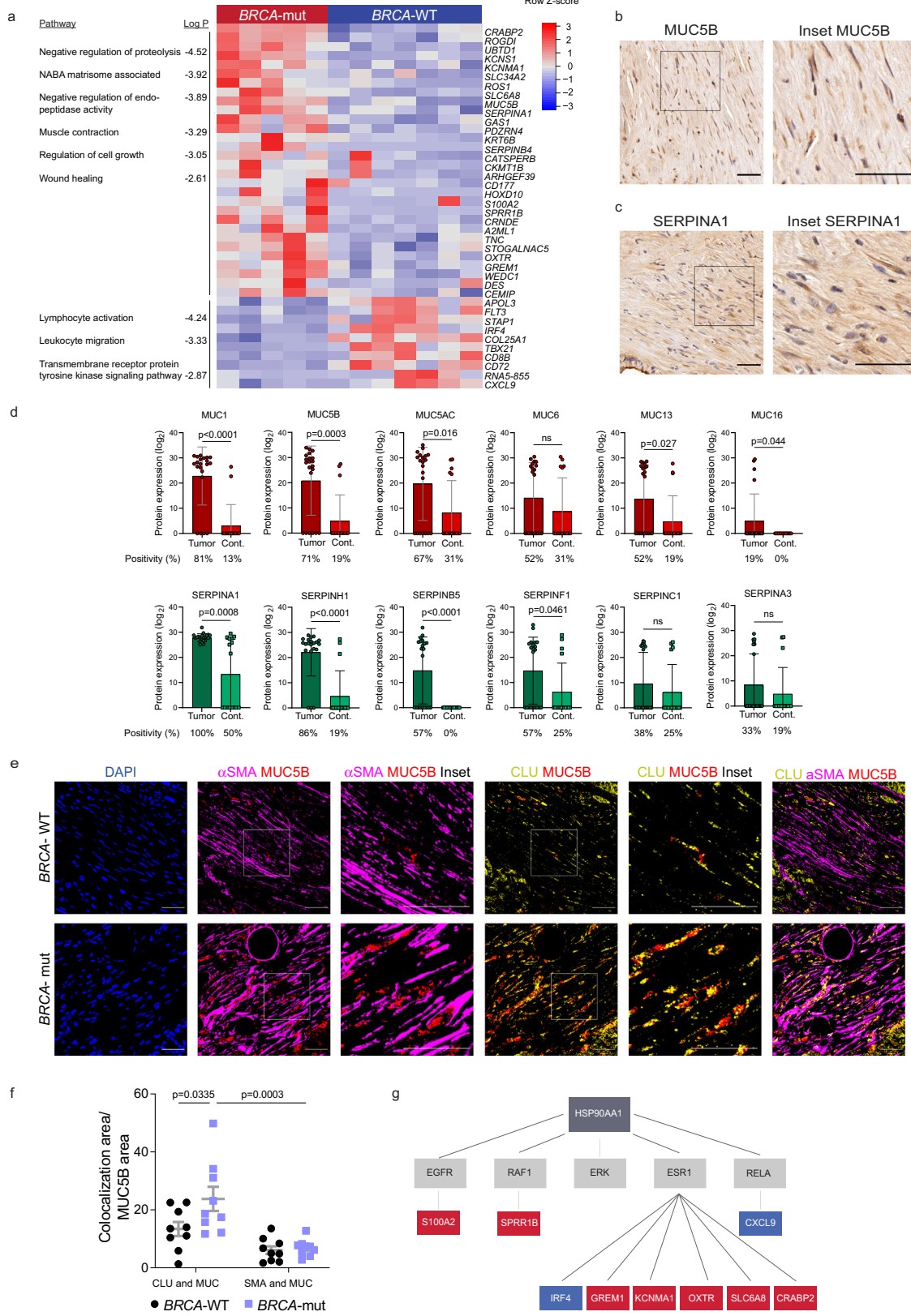

secreted proteins, which were previously proposed to serve as prognostic biomarkers of pancreatic neoplasms based on proteomic analysis of pancreatic fluids. SERPINA1 levels were elevated in PanIN3 lesions[58] and correlated to CLU expression in two lung cancer cell lines[59], and MUC5B was identified in pancreatic main duct fluid collected at the time of surgical resection[60] but no known association with

CLU was reported. IHC staining of tumor sections from PDAC patients showed that MUC5B and SERPINA1 are expressed by PDAC stromal cells (Fig. 2b, c; as well as by cancer cells; see Fig. 2e below). To test whether these proteins are secreted by PDAC human tumors, we assessed the exosomal content of 21 PDAC specimens and 16 normal adjacent controls in an independent patient cohort[61] (see Methods and

**Fig. 2 | The transcriptional profile of *BRCA*-mut stroma is different than that of *BRCA*-WT stroma.** CAF-rich regions of fresh-frozen tumor sections from 7 *BRCA*-WT and 5 *BRCA*-mut PDAC patients were laser-capture-microdissected and analyzed by RNA-seq. (**a**) Heatmap showing hierarchical clustering of DE genes in CAF-rich regions from *BRCA*-mut and *BRCA*-WT samples. Pathway analysis was performed using Metascape; Selected significant pathways ($p < 0.05$) are shown (see full list in Supplementary Data 4). (**b**, **c**) FFPE tumor sections from 2 PDAC *BRCA*-mut patients were stained by IHC for MUC5B and SERPINA1. Representative images are shown. Scale bar, 50 µm (**d**) Human PDAC tissue-derived exosomal proteomes ($n = 21$) and non-tumor adjacent tissue-derived exosomal proteomes ($n = 16$) were analyzed by liquid chromatography with tandem mass spectrometry (LC-MS/MS). Proteins found in >15% of pancreatic cancer exosomes were compared to pancreatic adjacent tissue-derived exosomes. Log2 protein expression of the indicated proteins is presented. *P* values were calculated by Welch's t-test for the comparison of expression level and Fisher's exact test for the comparison of positivity. Data are expressed as mean ± SEM. (**e**–**f**) FFPE tumor sections from 9 *BRCA*-mut and 9 *BRCA*-WT PDAC patients were stained by MxIF using antibodies for the indicated proteins. DAPI was used to stain nuclei. Scale bar, 50 µm. Representative images are shown in (**e**). MUC5B and SERPINA1 protein levels were quantified by ImageJ software and the area stained by each protein and CAF marker was measured. Quantification of MUC5B colocalization with CLU and αSMA was analyzed by two-way ANOVA, and presented as mean ± SEM in (**f**). (**g**) DE genes were analyzed by Ingenuity software using the causal network tool. Schematic representation of the predicted network is presented. Upregulated and downregulated genes in *BRCA*-mut patients are marked in red and blue, respectively; predicted regulators are marked in grey. Source data are provided as a Source Data file.

Supplementary Data 6). We detected multiple mucin and serpin proteins that were highly expressed in tumor exosomes compared to normal adjacent tissue-derived exosomes (Fig. 2d and Supplementary Data 6). Specifically, MUC5B was detectable in 71% of PDAC-derived exosomes, compared to 19% of adjacent pancreatic tissue-derived exosomes. SERPINA1 was found in 100% of PDAC-derived exosomes, however, it was also found in 50% of the control tissues (Fig. 2d and Supplementary Data 6).

The exosome cohort did not include the *BRCA* status, therefore we could not compare the exosomal levels of MUC5B and SERPINA1 in *BRCA*-mut *vs. BRCA*-WT tumors. Instead, we performed MxIF staining of SERPINA1 and MUC5B in *BRCA*-mut *vs. BRCA*-WT tumors. The total protein levels of SERPINA1 and MUC5B (when analyzing all stromal cells together) were similar in *BRCA*-mut *vs. BRCA*-WT. However, further analysis of MUC5B expression within the different CAF subtypes revealed significantly higher levels of MUC5B in CLU+ CAFs than in SMA+ CAFs. Moreover, the localization of MUC5B within CLU+ CAFs was significantly higher in *BRCA*-mut patients compared to *BRCA*-WT patients, suggesting a possible association between MUC5B and CLU+ CAFs (Fig. 2e, f and Supplementary Figure 2e).

To identify potential upstream regulators of the *BRCA*-associated CAF transcriptional program we analyzed our DE gene dataset using the Causal Network tool in the Ingenuity Pathway Analysis (IPA) software (see Methods)[62]. This analysis highlighted heat shock protein 90α gene, *HSP90AA1*, as a potential upstream regulator of multiple genes in our network (Fig. 2g). HSP90α is a stress-induced chaperone. Previous studies have reported a role for HSP90 in PDAC progression[63], and synergistic effects of CLU and HSP90α in promoting epithelial-to-mesenchymal transition and metastasis in breast cancer[64]. As both *CLU* and *HSP90AA1* are regulated by HSF1[65,66], the master transcriptional regulator of the heat shock response, we hypothesized that HSF1 may be orchestrating these *BRCA*-mut-induced transcriptional changes in the stroma.

## Activation of stromal HSF1 is elevated in *BRCA*-mut PDAC tumors

Work by us and others has shown indispensable roles for HSF1 in transcriptional rewiring of fibroblasts into CAFs in various cancer types[32–36]. To test whether HSF1 is differentially activated in *BRCA*-mut *vs. BRCA*-WT CAFs, we performed MxIF staining. HSF1 translocates to the nucleus upon activation, and thus its nuclear localization serves as a proxy for its activation (Fig. 3a, b; Supplementary Figure 3a). Comparing 14 *BRCA*-mut tumors with 20 *BRCA*-WT tumors from our patient cohort, we found significantly higher activation of HSF1 in *BRCA*-mut stroma compared to *BRCA*-WT stroma (Fig. 3c).

We were next curious to see whether other stress responses are also activated in *BRCA*-mut stroma, possibly due to DNA-damage-induced stress, or whether this phenomenon was specific to HSF1. To portray the stress network in PDAC, we stained for five additional stress-induced transcription factors (TFs): X-box binding protein 1 (XBP1)[67] and Activating Transcription Factor 6 (ATF6[68]; ER-stress

response); Hypoxia-inducible factor 1-α (HIF1α; Hypoxia)[30], Nuclear factor erythroid-2-related factor 2 (NRF2; oxidative stress)[69], and Activating Transcription Factor 4 (ATF4[70]; the integrated stress response; Fig. 3a, b). While none of these additional stress-activated TFs showed significant differential activation (Supplementary Figure 3a–f), a significant crosstalk between all these stress pathways was evident. All pairs of stress-TFs exhibited higher co-activation (per-patient) in *BRCA*-mut tumors compared to *BRCA*-WT tumors (Fig. 3d, e), suggesting that the stress inflicted by *BRCA* mutations is different than that found in a *BRCA*-WT PDAC microenvironment, leading to coordinated activation of a network of stress responses in the stroma of *BRCA*-mutated PDAC.

## HSF1 upregulates CLU/αSMA ratio in *BRCA*-mut tumors

CLU is an extracellular chaperone transcriptionally regulated by HSF1 in various contexts[65,71,72] and upregulated in response to DNA damage[73,74]. CLU was shown to play a critical role in promoting pancreas regeneration and tumorigenesis[75,76]. Supported by our findings of higher HSF1 activation and CLU/αSMA ratios in *BRCA*-mut tumors, we hypothesized that HSF1 may affect *BRCA*-associated CAF compositions through transcriptional regulation of stromal gene expression and specifically the regulation of *CLU* expression. To test this hypothesis, we first assessed the correlation between HSF1 activation and CLU/αSMA ratio in our clinical cohort. We found that HSF1 activation is correlated with CLU/αSMA ratio only in *BRCA*-mut patients and not in *BRCA*-WT patients (Fig. 4a, b). Next, we asked whether *CLU* expression is HSF1-dependent. To this end, we measured mRNA expression of *Clu* in primary PSCs isolated from WT and *Hsf1* null mice (Fig. 4c). We found that the expression of *Clu* was significantly lower in *Hsf1* null PSCs compared to WT PSCs, while the expression of other CAF markers, such as *Acta2* and *Il6*, was not altered (Fig. 4e, f). *Muc5b* showed somewhat reduced expression but this result was not significant (Fig. 4d).

To characterize the effect of *BRCA* mutations on HSF1-dependent *Clu* upregulation, we employed shRNA for *Brca2* in KPC cells (mimicking *BRCA2* loss-of-function) or non-targeting control (shControl; Supplementary Figure 4a). We chose to target *Brca2* rather than *Brca1* since mutations in *BRCA2* are more prevalent than in *BRCA1* in PDAC, and were found in 73% of our *BRCA*-mut cohort. Immortalized PSCs were cultured in 3D matrigel domes for four days in growth medium and four additional days in the presence of conditioned medium (CM) from KPC pancreatic cancer cell-organoids transduced with sh*Brca2* or shControl (Supplementary Figure 4b). Normal growth medium served as control for both conditions (Fig. 4g–j). These growth conditions were previously shown to suppress the myCAF phenotype and induce an inflammatory CAF phenotype (Supplementary Figure 4c–e and[13]). Indeed, we found that *Acta2* expression was abolished by the addition of KPC CM (Fig. 4i). In stark contrast, the expression of *Clu* and *Muc5b* was induced by addition of KPC CM, and silencing of *Brca2* in the KPCs led to a further, significant induction of both *Clu* and *Muc5b* expression (as compared to CM from KPC-

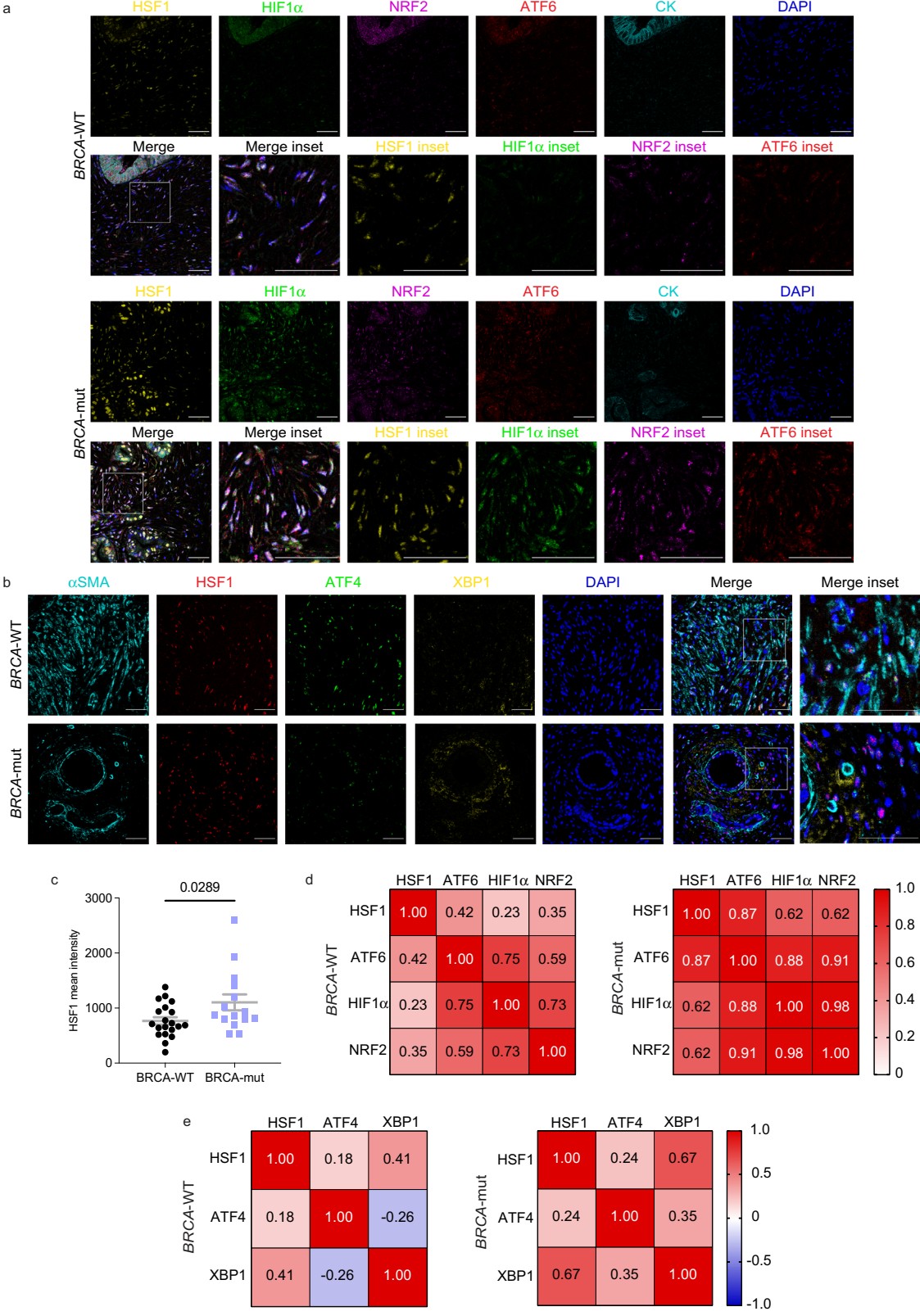

**Fig. 3 | A network of stress responses is activated in *BRCA*-mut stroma.** FFPE tumor sections from *BRCA*-mut and *BRCA*-WT PDAC patients were stained by MxIF using antibodies for the indicated proteins. (**a**, **b**) Representative images are shown. DAPI was used to stain nuclei. Scale bar, 50 μm. (**c**) Quantification of HSF1 mean intensity within all stromal cells in *BRCA*-mut (n = 14) and *BRCA*-WT samples (n = 20). 3-5 images per patient were analyzed using ImageJ software, HSF1 staining intensity was averaged within patients, and is presented as mean (across patients) ± SEM. Statistical analysis was conducted with a two-tailed unpaired t-test. (**d**, **e**) Pearson correlation matrices of stress TF coactivation in *BRCA*-mut (n = 8 for d; n = 13 for e) *vs*. *BRCA*-WT (n = 11 for d; n = 17 for e) patients. Source data are provided as a Source Data file.

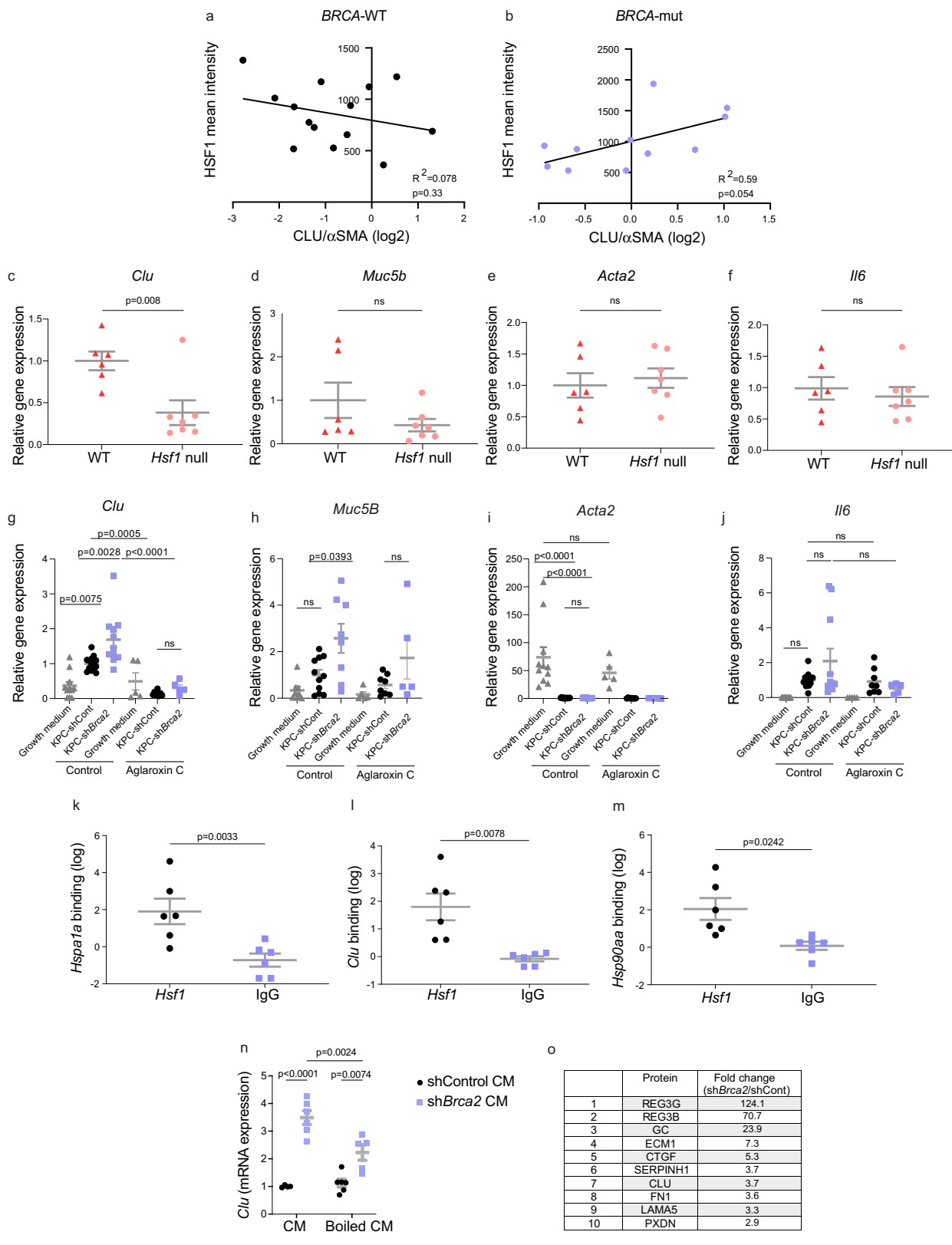

shControl or from PSCs; Fig. 4g, h). *Il6* expression was also induced by KPC CM, though this induction was not statistically significant (Fig. 4j). To test whether the cancer-induced upregulation of *Clu* and *Muc5b* is HSF1-dependent, we added to these cultures the synthetic small molecule CMLD011866 ((-)-aglaroxin C)[77–79]. This compound is a pyrimidinone variant of the rocaglate/flavagline natural product class, recently shown by us to inhibit HSF1 activity[33,79]. Aglaroxin C was added

to the CM every two days and the expression of *Clu, Muc5B, Acta2, and Il6* was measured (Fig. 4g–j). Treatment with aglaroxin C abolished the induction of *Clu* expression, suggesting that this induction is HSF1 dependent, and that HSF1 regulates the expression of this gene.

To examine if *Clu* and *Hsp90aa* are direct target genes of HSF1 in our system, we exposed PSCs to KPC-CM and performed chromatin immunoprecipitation (ChIP) with anti-HSF1 antibodies followed by

**Fig. 4 | HSF1 directly regulates *Clu* expression in *BRCA*-mut tumors.** (a–b) FFPE tumor sections from 14 *BRCA*-WT and 11 *BRCA*-mut PDAC patients were stained by MxIF for HSF1, CLU and αSMA. Images were analyzed by ImageJ. Pearson correlation between HSF1 mean intensity and CLU/αSMA ratio in the stroma of (a) *BRCA*-WT patients, and (b) *BRCA*-mut patients was calculated. (c–f) Expression levels of *Clu* (c), *Muc5b* (d) *Acta2* (e) and *Il6* (f) in primary PSCs freshly isolated from WT and *Hsf1*-null mice. n = 6 WT and 7 *Hsf1*-null mice, combined from 3 independent experiments. Statistical analysis was conducted via unpaired two-tailed t-test (g–j) Immortalized PSCs were seeded in Matrigel for 4 days. Conditioned media (CM) from KPC cells in which *Brca2* was silenced by shRNA or non-targeting shControl (KPC) was then added for an additional 4 days, or cells were left in growth medium as control. 3 nM of the HSF1 inhibitor, CMLD011866 (aglaroxin C), or PBS control was added to the conditioned media (CM) every 2 days, for 4 days, after which the expression levels of *Clu* (g), *Muc5b* (h) *Acta2* (i), and *Il6* (j), were measured by qRT-PCR. For each condition n = 5-14 biologically independent culture domes combined from 3 independent experiments. Statistical analyses were conducted using one way ANOVA and Tukey' test for multiple comparisons. (k–m) Immortalized PSCs were cultured with or without KPC-shControl-CM for 24 h. ChIP-PCR was performed for putative heat-shock elements of *Hsp1a1* (k), *Clu* (l), and *Hsp90aa* (m), and for an intergenic region for normalization, following pulldown with anti-HSF1 antibody compared to IgG control. One-way ANOVA was performed on log2 values to compare between the group ratios of expression and Tukey's test was performed to adjust for multiple comparisons. Data are presented as mean ± SEM for each primer normalized to the intergenic control in (c–m) (n = 6 biologically independent culture wells combined from 3 independent experiments). (n) PSCs were cultured in the presence of boiled or unboiled CM from KPC-shControl (unboiled n = 4, boiled n = 6 biologically independent samples) and KPC-sh*Brca2* organoids (unboiled n = 6, boiled n = 5 biologically independent samples) as described in (g–j). Expression of *Clu* in PSCs were subsequently measured by qRT-PCR. Statistical analysis was conducted with two-way ANOVA. Data are presented as mean ± SEM (o) Top 10 differentially expressed proteins (fold change of protein abundance based on mass-spectometry label-free quantification, sh*Brca2*/shControl) from CM of KPC-shControl and KPC-sh*Brca2* organoids as measured by mass-spectrometry. Source data are provided as a Source Data file.

qPCR with primers flanking heat-shock elements on the DNA of *Clu and Hsp90aa*. *Hspa1a*, a well-known HSF1-target gene, served as control (Fig. 4k–m). Both *Clu* and *Hsp90aa* were significantly enriched in the HSF1-bound fraction compared to IgG control, demonstrating direct regulation of these genes by HSF1 in cancer-conditioned PSCs (Fig. 4k–m). Together, these findings suggest that *BRCA*-deficient cancer cells induce an HSF1-dependent transcriptional program in PSCs.

In search for factors that may mediate this effect, we next analyzed the medium conditioned by KPC-sh*Brca2* cells. First, to test whether HSF1 and *Clu* upregulation is mediated by secretion of proteins, we boiled CM from KPC-shControl and KPC-sh*Brca2* organoids and treated PSCs with either unboiled or boiled CM. Boiling of KPC-sh*Brca2* CM significantly reduced expression of *Clu* by PSCs (Fig. 4n), suggesting that this effect is mediated by a secreted protein(s). Next, we preformed mass-spectrometry analysis of the organoid CM. The two most differentially secreted proteins from KPC-sh*Brca2* relative to KPC-shControl organoids were the Regenerating islet-derived (Reg) proteins, REG3B/G (Fig. 4o, Supplementary Data 7). REG3B/G are C-type secreted lectins that play active roles in pancreatitis and in the transition from pancreatitis to pancreatic cancer through different mechanisms, including induction of STAT3, RAF-MEK-ERK signaling, and immune cell modulation[80–83]. Of note, we have previously demonstrated an association between REG3B/G and HSF1 signaling, by showing that REG3B/G are upregulated during inflammation in the colon in an HSF1-dependent manner[33]. In fact, six of the ten most differentially secreted proteins from KPC-sh*Brca2* *vs* KPC-shControl organoids were previously shown by us to be upregulated during colon inflammation in an HSF1-dependent manner (REG3G, REG3B, GC, SERPINH1, FN1, and PXDN)[33]. We also found CLU itself in this list. These findings suggest that loss of BRCA2 in cancer cells leads to differential secretion of proteins resulting activation of HSF1 in stromal fibroblasts, and, potentially, also in the cancer cells themselves.

### *BRCA*-deficient cancer cells induce a distinct transcriptional program in PSCs

To further dissect the transcriptional shift induced by *BRCA*-deficient cancer cells in PSCs we performed RNA-seq of PSCs following 3D Matrigel cultures in the presence of KPC-sh*Brca2*-CM, KPC-shControl-CM, or normal growth medium (DMEM), as control (Fig. 5a, Supplementary Data 8 and Supplementary Figure 5a–c). In parallel, we sequenced KPC-sh*Brca2* and KPC-shControl cells (Supplementary Figure 5d–f and Supplementary Data 9), which confirmed that *Brca2* is among the top 20 differentially downregulated genes in KPC-sh*Brca2* *vs* KPC-shControl (Supplementary Figure 5f). In addition to the shared response to cancer CM, we detected distinct transcriptional changes in PSCs exposed to KPC-sh*Brca2* CM *vs* KPC-shControl CM

(Supplementary Data 8). 31 genes were differentially upregulated only by KPC-sh*Brca2* CM, and not by KPC-shControl CM. Notably, *Clu* ranked 6th on this list, highlighting its prominence in *Brca2*-mut reprogramming of PSCs (Fig. 5a and Supplementary Data 8).

PSCs are highly plastic and assume distinct transcriptional and functional properties depending on culture conditions (2D *vs* 3D, cancer CM etc)[11]. To explore the transcriptional changes of CAFs in an in-vivo setting, we inoculated KPC-sh*Brca2* and KPC-shControl cells orthotopically into the pancreata of C57BL/6 J mice. Three weeks later, tumors were harvested, digested into single-cell suspensions, and CAFs were isolated by fluorescence-activated cell sorting (FACS; Fig. 5b and Supplementary Figure 5g). The cells were lysed immediately after sorting and processed for RNA-seq. Differential expression analysis revealed 482 genes significantly upregulated and 666 genes significantly downregulated in CAFs from KPC-sh*Brca2* tumors compared to CAFs from KPC-shControl tumors (Fig. 5c, d, Supplementary Figure 5h, and Supplementary Data 10). Pathway analysis highlighted cell adhesion, MAPK-cascade regulation, and positive regulation of cell death among the most differentially upregulated pathways in CAFs from KPC-sh*Brca2* tumors compared to those from KPC-shControl tumors (Supplementary Data 10). ECM organization, IGF signaling and TGFβ signaling were differentially downregulated in these CAFs compared to CAFs from KPC-shControl tumors (Fig. 5c, d and Supplementary Data 10). *Clu* was among the significantly upregulated in CAFs from KPC- sh*Brca2* tumors compared to KPC-shControl tumors (Supplementary Data 10), consistently supporting its activation in *BRCA*-mut human tumors and *Brca2*-deficientcancer-conditioned PSCs.

Our finding that HSF1 is preferentially activated in CAFs of *BRCA*-mutated tumors, and preferentially induces the expression of *Clu* in sh*Brca2*-conditioned-PSCs and CAFs, suggested that HSF1 may serve as a master regulator of the *BRCA*-CM-mediated transcriptional shift. To test this, we systematically queried a publicly-available dataset of HSF1 target genes (https://hsf1base.org/)[84] for the DE genes between PSCs conditioned by KPC-sh*Brca2*-CM and KPC-shControl-CM (Fig. 5e). We performed a similar search also for the DE genes between CAFs from KPC-sh*Brca2* tumors *vs* CAFs from KPC-shControl tumors (Fig. 5f). We found that HSF1 targets were significantly enriched in genes upregulated in *Brca2*-associated PSCs and CAFs (Fig. 5e, f and Supplementary Data 11), supporting the notion that HSF1 plays a role in regulating the *BRCA*-deficiency-mediated stromal transcriptional shift.

### PSCs reprogrammed by *Brca2*-deficient cancer cells shift from myofibroblastic into immune-regulatory CAFs

The shift in CLU⁺/SMA⁺ CAF ratios in human patients, and the shift in *Clu vs. Acta2* expression in PSCs conditioned by *Brca2*-deficient cancer cells, suggest that CAFs of *BRCA*-mutated tumors may undergo a shift from myofibroblastic to immune-regulatory functions. Supporting this

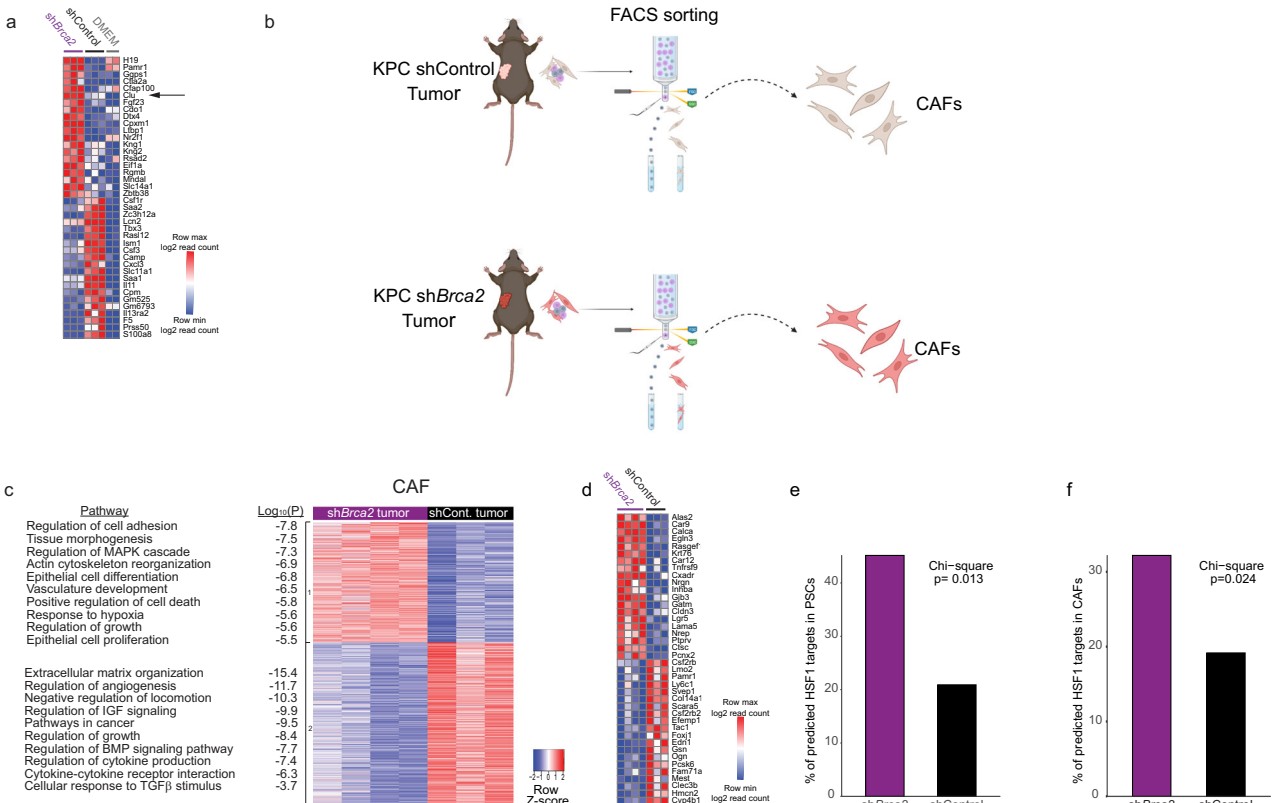

**Fig. 5 | BRCA2 expression in the cancer cells affects the transcriptional profile of PSCs and CAFs.** (**a**) A heatmap representing the 20 most upregulated genes in KPC-sh*Brca2*-CM-treated PSCs compared to shControl, (see Methods and Supplementary Data 8, tab 7), and the 20 most upregulated genes in KPC-shControl-CM-treated PSCs (compared to sh*Brca2*). *Clu* is marked by an arrow. $n = 2$ biologically independent domes for DMEM treated PSCs and $n = 3$ biologically independent domes for shControl-CM-treated and KPC-sh*Brca2*-CM-treated PSCs. (**b**) Schematic representation of the workflow. $2 \times 10^4$ KPC-sh*Brca2* or KPC-shControl cells were inoculated into the pancreata of syngeneic C57BL/6 J mice. Three weeks later the mice were sacrificed, tumors were dissected and CAFs were isolated by FACS sorting (see Methods). The scheme was generated with biorender.com. (**c, d**) RNA-seq analysis of CAFs from KPC tumors. $n = 3$ mice for KPC-shControl tumors

and $n = 4$ mice for KPC-sh*Brca2* tumors (**c**) Heatmap representing hierarchical clustering of the DE genes between CAFs from KPC-sh*Brca2* or KPC-shControl tumors (right), and pathways enriched in each cluster with their corresponding p-values (left). Pathway analysis was performed using Metascape, selected pathways are shown, see Supplementary Data 10 for the full list. (**d**) A heatmap representing the 20 most upregulated genes in KPC-sh*Brca2*-tumor CAFs, and the 20 most upregulated genes in KPC-shControl-tumor CAFs. (**e, f**) DE genes between KPC-sh*Brca2*-CM treated- and KPC-shControl-CM treated-PSCs (**e**) and CAFs (**f**) were matched to a database of murine HSF1 targets genes. The statistical significance of the dependency between treatment (sh*Brca2*/ shControl) and HSF1 targets was tested using a Chi-square test. Source data are provided as a Source Data file.

notion, *Pdl1*, whose upregulation in PDAC CAFs was suggested to mediate T-cell immune suppression[19,85,86], was induced in PSCs by KPC-sh*Brca2* CM, but not by KPC-shControl CM (Fig. 6a). Similar to *Clu*, the induction of *Pdl1* was inhibited by aglaroxin C, suggesting an increased immune-regulatory function for PSCs induced by *BRCA*-deficient cancer cells, in an HSF1-dependent manner (Fig. 6a). To functionally test the ability of CAFs isolated from *Brca2*-deficient tumors to regulate immune cell function, we isolated CD8+ T-cells from spleens of naïve C57BL/6 J mice, and activated them in the presence of CAFs from KPC-shControl or KPC-sh*Brca2* tumors (Fig. 6b, c and Supplementary Figure 6a, b). T-cells activated in the presence of CAFs from *Brca2*-deficient tumors were significantly more repressed in their ability to upregulate the activation markers CD25 and CD69 compared to T-cells activated in the presence of CAFs from shControl tumors.

To search for potential factors that could be mediating the inhibitory effect of CAFs from sh*Brca2* tumors on T-cells, we mined our tumor-derived CAF RNA-seq data from KPC tumors using the ICELL-NET receptor-ligand analysis tool employing a CD8+ T-cell-receptor dataset[87]. ICELLNET is a computational tool that calculates a communication score for ligand-receptor interactions based on transcriptomic data and a database of potential ligand-receptor pairs. We applied ICELLNET to screen for immune modulatory surface ligands in CAFs that may inhibit T-cell activity, and found that CAFs from KPC-

sh*Brca2* tumors scored higher than KPC-shControl CAFs in the checkpoint signaling axis involving the TIGIT and CD96 T-cell inhibitory receptors, and their cognate ligands CD155 and Nectin1-3 (Fig. 6d). Notably, *Nectin2* (PVRL2) is also differentially upregulated in *CLU*high CAFs in human PDAC as shown by our analysis of scRNA-seq data from Peng et al. (Supplementary Data 2). To experimentally validate the cell-surface expression of inhibitory checkpoint markers on CAFs, we isolated CAFs from KPC-shControl or KPC-sh*Brca2* tumors, and performed FACS analysis using antibodies for PD-L1, CD155, and Nectin2. CAFs from KPC-sh*Brca2* tumors demonstrated higher cell surface expression of CD155 and Nectin2 (as compared to KPC-shControl), and a similar trend was observed for the expression of PD-L1 (Fig. 6e–g and supplementary Fig. 6c), confirming the ICELLNET receptor-ligand results and further supporting their role in suppressing T-cell function. These findings also further strengthen the connection between *Clu* upregulation and induction of immune regulatory pathways in CAFs.

Next, we assessed myofibroblastic functions. To that end, we measured the ability of PSCs to secrete collagen, in-vitro, using Sirius Red staining. We found that PSCs conditioned by KPC-sh*Brca2* medium secreted significantly less collagen than PSCs conditioned by KPC-shControl medium (Fig. 6h, i). To test the relevance of these findings in patients we assessed ECM organization in the vicinity of CLU+ or αSMA+ CAFs by second harmonic generation (SHG) imaging combined with

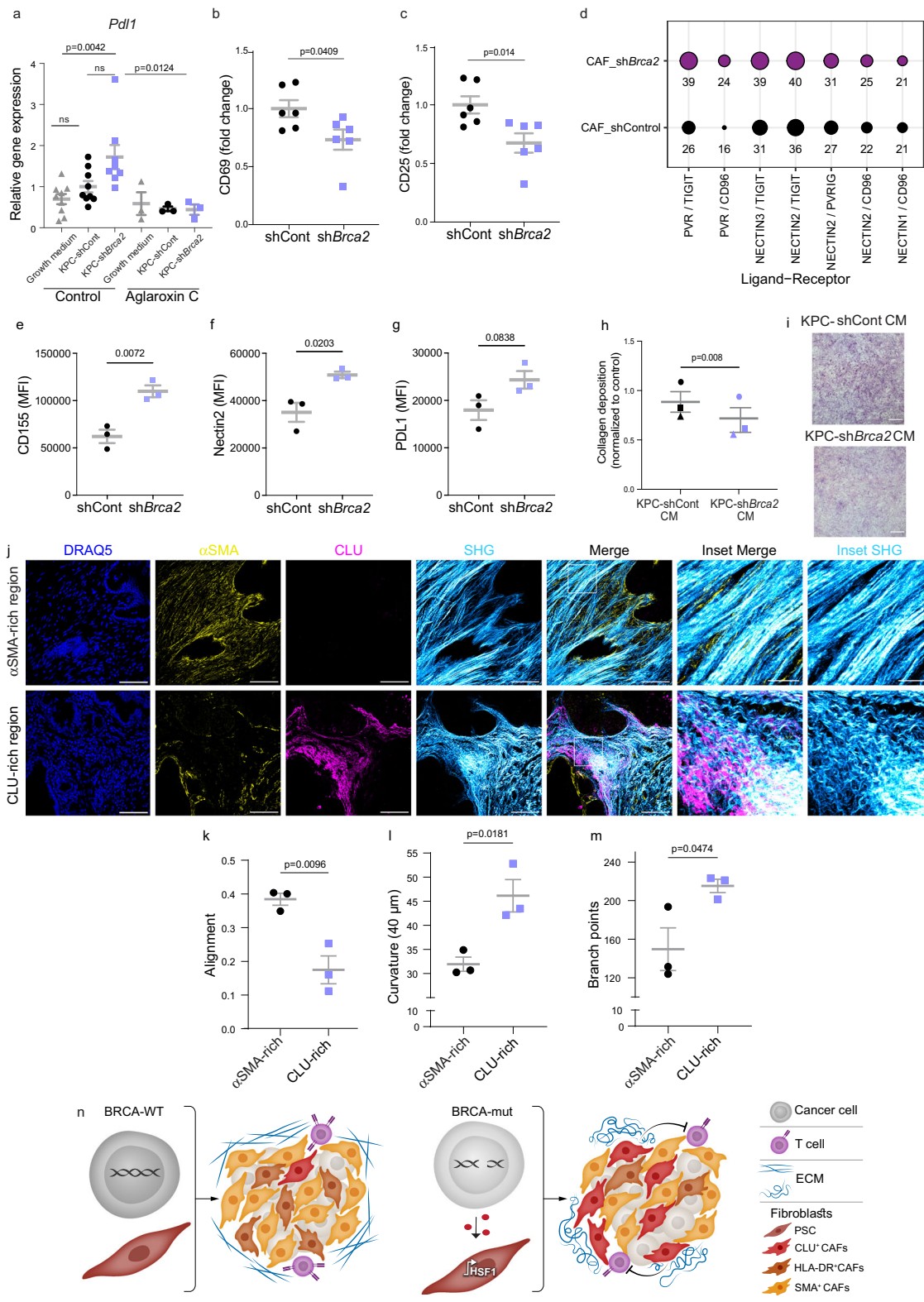

MxIF staining in *BRCA*-WT *vs BRCA*-mut patients (Fig. 6j–m). CLU-rich stromal regions that are abundant in *BRCA*-mut tumors demonstrated an altered ECM architecture, characterized by significantly reduced parallel alignment, and increased curvature and branching of collagen streaks compared to αSMA-rich regions (Fig. 6j–m).

Overall, our findings suggest that *BRCA*-mut cancer cells promote a stressful TME that leads to the activation of HSF1 in a subset of PSCs. These PSCs are reprogrammed into immune regulatory CLU⁺ CAFs,

resulting in a different stromal landscape in *BRCA*-mut compared to *BRCA*-WT PDAC tumors (Fig. 6n).

## Discussion

Accumulating evidence over the past few years unraveled vast heterogeneity of CAFs in the TME[7,8,88,89]. This heterogeneity was proposed to stem from different cells of origin giving rise to CAFs[37–41,90], and from transcriptional rewiring driven by different external cues received

**Fig. 6 | *Brca2*-deficient cancer cells shift CAF functions. (a)** Immortalized PSCs were seeded in Matrigel for 4 days. CM from PSCs or from KPC cells in which *Brca2* was silenced by shRNA or nontargeting shControl (KPC) was then added for an additional 4 days. 3 nM of aglaroxin C or PBS control was added to the conditioned media. *Pdl1* was measured by qRT-PCR. Statistical analysis was conducted via two way ANOVA. Data are presented as mean ± SEM. For each condition $n = 3$-9 biologically independent culture domes combined from 3 independent experiments. **(b–c)** CD8[+] T-cells were isolated from spleens of naïve C57BL/6 J mice and activated in the presence of CAFs isolated from either KPC-shControl or KPC-sh*Brca2* tumors. Subsequently, T-cells were subjected to FACS analysis for surface expression of CD69 and CD25. Statistical analysis was conducted via unpaired two tailed t test. Data quantified are presented as mean ± SEM of 6 KPC-shControl and 6 KPC-sh*Brca2* mice. **(d)** Scores of ligand-receptor binding were calculated using the ICELLNET R package (see Methods) to predict potential differential interactions between ligands of CAFs derived from shControl *vs* sh*Brca2* tumors with immune checkpoint receptors on CD8[+] T-cells. **(e–g)** CAFs were isolated from KPC-shControl ($n = 3$ mice) and sh*Brca2* ($n = 3$ mice) and stained with anti-CD155 (**e**), anti-Nectin2 (**f**), and anti-PD-L1 (**g**). Statistical significance was assessed by two tailed t-test. Data are presented as mean ± SEM **(h-i)** PSCs were treated with CM derived from KPC-sh*Brca2* or KPC-shControl organoids, or with growth medium as control for 4 days. Then, cultures were stained with Sirius red (SR) to assess collagen

deposition (see Methods). Each point represents the average of sh*Brca2* or shControl normalized to the average of the growth medium control in each experiment. $n = 3$ independent experiments, each representing an average of 5 independent culture wells. Two tailed t-test was performed on normalized values. **(i)** Representative images of PSCs stained with SR following 4 days treatment with CM derived from KPC-sh*Brca2* or KPC-shControl organoids. Scale bar, 300 μm. **(j–m)** FFPE tumor sections from *BRCA*-mut and *BRCA*-WT PDAC patients were stained by double staining for αSMA and CLU and imaged using Second harmonic generation (SHG) imaging. **(j)** Representative images are shown. DRAQ5 was used to stain nuclei. Scale bar, 100 μm, or 25 μm (inset). **(k–m)** Quantification of matrix pattern using the TWOMBLI plug-in (see Methods) ($n = 3$ BRCA-WT and $n = 3$ *BRCA*-mut patients). The following parameters were analyzed: **(k)** Alignment−the extent to which fibers within the field of view are oriented in a similar direction; **(l)** Curvature- the mean change in angle moving incrementally along 40 μm mask fibers; and **(m)** Branchpoint−the number of intersections of mask fibers in the image. Statistical analysis was conducted via unpaired two tailed t test. Data are presented as mean ± SEM **(n)** Schematic representation of the proposed model. Secreted factors from *BRCA*-mutated cancer cells induce *HSF1* activation in a subset of adjacent PSCs leading to their transcriptional rewiring into immune-regulatory CLU[+] CAFs. Source data are provided as a Source Data file.

from neighboring cells and local environmental conditions[28,29]. The contribution of germline mutations in the cancer cells to stromal rewiring is largely uncharted. Here we show that *BRCA* mutations in the cancer cells elicit stromal reprogramming in the microenvironment resulting in distinct stromal landscapes of *BRCA*-mut PDAC compared to *BRCA*-WT PDAC. Specifically, we show that human *BRCA*-mut tumors express higher levels of CLU[+] CAFs, and, consequently, higher CLU[+]/αSMA[+] and CLU[+]/HLA-DR[+] CAF ratios. We portray the transcriptional landscapes and potential upstream regulators of CAFs in *BRCA*-mut and *BRCA*-WT tumors from patients, and reveal a network of stress responses activated in *BRCA*-mut-associated stroma. Within this network, we find a specific role for HSF1 as the transcriptional regulator of *Clu*. Using cancer organoids, co-cultures, and in-vivo models we show that HSF1 mediates a transcriptional shift of PSCs into CLU[+] CAFs which exert immune regulatory characteristics (Fig. 6n).

Recent studies by us and others have utilized single-cell RNA-sequencing and imaging technologies to classify CAFs into functional subtypes based on differential expression of cell surface markers and genes. Here we find three CAF subtypes, distinctively marked by αSMA, CLU and HLA-DR (i.e. MHC-II). αSMA is a classic marker for myofibroblasts. MHC-II was only recently discovered to mark a subtype of CAFs[8,12], referred to as antigen-presenting CAFs, though their actual antigen-presenting activities remain to be elucidated. CLU marks SMA[low] CAFs in mouse models of breast and pancreatic cancer[8,13,40]. Our MxIF analysis showing that CLU marks a discrete population from HLA-DR[+] CAFs, together with the analysis of human scRNA-seq datasets highlighting immune-regulatory pathways in *CLU*[+] CAFs, suggest that in human PDAC, CLU marks immune-regulatory CAFs. These CLU[+] CAFs are significantly upregulated in *BRCA*-mut PDAC compared to *BRCA*-WT. Our co-culture and mouse studies confirm these findings from human patients and suggest that in mice too, immune-regulatory CLU[+] CAFs are significantly upregulated in *BRCA*-deficient PDAC compared to *BRCA*-WT. *BRCA*-mut PDAC immune microenvironments in other cancer types are characterized by increased infiltration of T cells[49–52]. Our IF analysis of CD3 staining in patients did not show increased T cell infiltration, however it does not exclude the possibility that the T cell composition is altered. Moreover, CLU[+] CAFs may act to modulate the activity of additional cells in the immune microenvironment such as macrophages.

*Clu* is transcriptionally regulated by the stress-induced master regulator, HSF1[65]. HSF1 has been shown by us and others to play key roles in the transcriptional rewiring of fibroblasts into CAFs in various cancers[32–36]. Here we describe a preferential activation in a specific

subtype of cancer, *BRCA*-mut PDAC, and expose a new facet of HSF1's stromal activities, affecting CAF composition. Preferentially activated in the stroma of *BRCA*-mut PDAC, HSF1 activates *Clu*, and leads to induction of immune-regulatory CLU[+] CAFs.

CLU is a molecular chaperone, harboring two isoforms−a nuclear isoform (nCLU) and a secreted one (sCLU). These isoforms were shown to have opposing activities; nCLU is a pro-apoptotic factor, while sCLU is a stress-induced, pro-survival factor. In epithelial cells, sCLU is upregulated by DNA damage[91], it is overexpressed in various cancers[92], and the shift of nCLU to sCLU expression is associated with progression towards high-grade and metastatic carcinoma in different cancers[93–96]. The nuclear-to-secreted transition of CLU has not been extensively characterized in fibroblasts. Here, we detect upregulation of the secreted form of CLU in CAFs of *BRCA*-mut PDAC. Moreover, silencing of *Brca2* in cancer cells is sufficient to induce *Clu* expression in CAFs of mouse tumors and in WT PSCs in culture, confirming not only that *Clu* is induced by *BRCA* deficiency but also that this is a non-cell-autonomous pathway induced by *BRCA* deficiency in the cancer cells. These CAFs have immune regulatory effects and modulate the adjacent ECM organization. Previous studies have implicated TGFβ in promoting the transition of CAFs from inflammatory to myofibroblast-like[14]. TGFβ was also shown to negatively regulate the expression of sCLU in fibroblasts during fibrosis[97,98]. Our RNA-seq analysis of LCM stroma from patients shows upregulation of the TGFβ inhibitor, Gremlin1 (*GREM1*), in *BRCA*-mut PDAC, and our RNA-seq analysis of mouse KPC and CAFs shows differential downregulation of TGFβ signaling both in KPC-sh*Brca2* cells and in CAFs from KPC-sh*Brca2* tumors compared to CAFs from KPC-shControl tumors (Fig. 5b and Supplementary Figure 5d). In line with our findings, GREM1[+] fibroblasts were previously shown to be upregulated during chronic pancreatitis and PDAC[99], possibly promoting disease progression through M2 macrophage polarization. Another recent study demonstrated that GREM1 orchestrates cellular heterogeneity in PDAC by maintaining the epithelial compartment[100]. Since cellular heterogeneity in PDAC is a prominent characteristic of PDAC subtype designation[101], GREM1 expression by CAFs and paracrine signaling with epithelial subpopulations may play an important role in maintenance of epithelial heterogeneity in *BRCA2* mutated tumors. Taken together these findings further support the notion that *BRCA* mutations in the cancer cells lead to a stromal shift from TGFβ induced SMA[+] myofibroblasts to HSF1-induced CLU[+] fibroblasts.

Clinical studies using inhibition of CLU by single-stranded antisense oligonucleotides showed elevated sensitivity to chemotherapy

and radiotherapy in cancer patients[102]. In addition, CLU was shown to regulate DNA repair pathways, including the *BRCA1* pathway[103]. These findings suggest that *BRCA*-mut patients may show improved response to CLU inhibition. Moreover, highlighted by our RNA-seq data, HSP90 is suggested to partially regulate the *BRCA*-driven transcriptional changes we identified. Notably, CLU was shown to form a complex with HSP90 proteins[104] to synergistically promote EMT and metastasis[64], and their combined inhibition showed improved response in prostate cancer[105]. Nevertheless, inhibiting the expression of CLU⁺ CAFs may shift the balance and enhance the relative expression of SMA⁺ CAFs, resulting in a stiff, myofibroblast-rich stroma that may be more pro-tumorigenic than the CLU-rich stroma[19]. Thus, future studies should test the outcome of CLU inhibition and/or of HSP90 inhibition in combination with platinum-based chemotherapy or PARP inhibitors on *BRCA*-mut cancer. Future studies should also assess the efficacy of combining immune-regulatory CAF inhibition with immunotherapies— while early phases of clinical trials show promising results in combining PARP-inhibitors with immunotherapy in *BRCA* mutated tumors, the role of *BRCA1/2* mutations in immunotherapy is still controversial[106,107]. How CAFs play into the response to immunotherapy in this context is still unknown.

This work identifies a unique stress-response network that is activated in *BRCA*-mut stroma. Tumors are stressful environments and stress responses are well known to play important roles in supporting survival of cancer cells. For example, activation of *Nrf2* in cancer cells leads to elevated mRNA translation and mitogenic signaling[69], the endoplasmic reticulum (ER) stress response was shown to mediate chemoresistance in PDAC cells[108], and expression of ATF4 in fibroblasts was suggested to promote disease progression and resistance to chemotherapy in PDAC[70]. Other studies reported regulation of stress responses by *BRCA1*. *BRCA1* was shown to actively regulate reactive-oxygen-species (ROS) in response to oxidative stress[109], and to regulate the unfolded protein response (UPR)/ER stress response by regulating glucose-regulated protein (GRP)78, CHOP and GRP94[110]. Furthermore, *BRCA1* induction led to downregulation of *HSF1*[111]. Our results indicate that while only HSF1 was significantly higher in *BRCA*-mut tumors compared to *BRCA*-WT tumors, a broader stress network is activated in the *BRCA*-mut TME. This may reflect a mechanism by which DNA repair deficiencies in the cancer cells impose unique stress conditions on the TME that reshape the stress-response network in the stroma. Nevertheless, HSF1 appears to play a dominant role in this network, perhaps also through activation in the cancer cells themselves, as reflected by our mass-spectrometry analysis of proteins secreted from KPC-sh*Brca2* cells. Indeed, HSF1 is well-known to be activated in cancer cells of various tumor types[112], and may be mediating a pro-tumorigenic feedback loop between cancer cells and CAFs in *BRCA2*-deficient tumors.

CAFs are highly plastic and dynamically shift between myofibroblastic and immune-regulatory functions when exposed to different microenvironments, including different culture conditions, making their functional characterization challenging[7]. Nevertheless, by combining organoid cultures and mouse injections of cancer cells with 3D cultures of PSCs and CAFs we could define a functional shift induced by *BRCA*-deficient cancer cells in a subset of CAFs. We find that PSCs conditioned by *BRCA*-deficient cancer cells exert reduced collagen production activity and increased immune-regulatory activity than PSCs conditioned by *BRCA*-proficient cancer cells. In patients, CLU⁺ CAFs associate with distinct ECM structures compared to SMA⁺ CAFs. Several recent single-cell studies performed by us and others on human tumors and mouse models identified diverse CAF subtypes in breast, pancreatic, ovarian and prostate cancers[8,10,22,37,113,114]. To which extent these subtypes are cancer-type specific or represent pan-cancer markers is a burning question in our field. Even more pressing is the question of whether the mutation dependencies identified in our study are PDAC-specific or Pan-*BRCA*, or perhaps even represent a general characteristic

of homologous recombination deficiency (HRD) cancers. These questions bear important implications on future therapeutic strategies. Defining common and segregating design principles of CAFs between tissues and organs sharing similar *BRCA*/HRD mutations will be an important step towards advancing therapy directed at these poor-prognosis cancers.

## Methods

### Ethics statement
All clinical samples and data were collected following approval by Memorial Sloan Kettering Cancer Center (MSKCC; IRB, protocols #06-107, #15-015 and 13-217), Shaare Zedek Medical Center (IRB protocol #101/13; Ministry of Health no. 920130134), Sheba Medical Center at Tel-Hashomer (IRB protocol #0967-14-SMC), and the Weizmann Institute of Science (IRB, protocols # 186-1) Institutional Review Boards. All animal studies were conducted in accordance with the regulations formulated by the Weizmann Institute of Science Institutional Animal Care and Use Committee (IACUC; protocol #02040220-2, #33870217-2, #32520117-2, #06920921-2).

### Human patient samples
Tumor samples from surgically resected primary pancreas ductal adenocarcinomas were from patients treated at Memorial Sloan Kettering Cancer Center (MSKCC), at Shaare Zedek Medical Center, and at Sheba Medical Center; informed consent to study the tissue was obtained via MSK IRB protocols #06-107 and 13-217 (Cohort 1; Supplementary Data 1), and #15-015 for the exosome analysis (Cohort 2; Supplementary Data 6), and via Shaare Zedek Medical Center (IRB protocol #101/13; Ministry of Health no. 920130134), Sheba Medical Center at Tel-Hashomer (IRB protocol #0967-14-SMC), and the Weizmann Institute of Science (IRB, protocols # 186-1) Institutional Review Boards. Cohort 1 included a total of 27 *BRCA*-WT PDAC patients and 15 *BRCA*-mut PDAC patients (Supplementary Data 1). Cohort 2 included fresh samples from 26 patients from which tumor tissues and/or normal adjacent controls were collected (Supplementary Data 6). Of the 15 *BRCA*-mut patients, 4 are *BRCA1*-mut carriers and 11 are *BRCA2* carriers, which is consistent with the reported prevalence of *BRCA1* and *BRCA2* mutations in PDAC[115]. FFPE whole tumor sections and deeply annotated demographic, clinical, pathologic and genomic (MSK-IMPACT™) data were collected for all MSKCC patients in the study. In addition, fresh-frozen tumor tissue was collected for a subset of 12 patients.

### Mice
C57BL/6 J 8-week males were purchased from Envigo (Jerusalem Israel). 8-week male *Hsf1* null mice and their WT littermates (BALB/c × 129SvEV, by Ivor J. Benjamin[116]) were maintained under specific-pathogen-free conditions at the Weizmann Institute's animal facility. Mice were sacrificed by $CO_2$ for pancreata harvesting. All animal studies were conducted in accordance with the regulations formulated by the Institutional Animal Care and Use Committee of the Weizmann Institute of Science (WIS) (IACUC; protocol #02040220-2, #33870217-2, #32520117-2, #06920921-2).

### Cell lines and primary cell cultures
**Mouse-immortalized PSCs, the KPC cell line and KPC organoids** were provided by David Tuveson's laboratory[13]. Immortalized mouse PSCs and the KPC cell line were cultured in growth medium containing Dulbecco's modified Eagle's medium (DMEM; Biological industries, 01-052-1 A) supplemented with 10% fetal bovine serum (FBS) and pen/strep. Silencing of *Brca2* in KPC cell lines was achieved by lentiviral infection, using mouse *Brca2* shRNA in pLKO.1 vector (Horizon, RMM4534) and pLKO.1 non-targeting control vector (Sigma Aldrich, SHC002), and puromycin selection.

                                          

**Primary PSC isolation.** Pancreata were collected postmortem from *Hsf1* null mice or WT littermates into HBSS (Sigma-Aldrich, H6648), then minced into Roswell Park Memorial Institute 1640 (RPMI) (Biological industries, 01-100-1 A), supplemented with 0.5 mg/mL Collagenase D (Merck, 11088866001), 0.1 mg/mL Deoxyribonuclease I (Worthington, LS002007) and 1 mM HEPES (Biological Industries, 03-025-1B). Pancreata were incubated at 37 °C for 40 min with mechanical disruption every 5 min. Cells were then filtered with 100 µm filters, centrifuged and isolated by Histodenz gradient (Sigma-Aldrich, D2158) dissolved in HBSS. Cells were resuspended in HBSS with 0.3% BSA and 43.75% Histodenz, HBSS with 0.3% BSA was layered on top of the cell suspension, and centrifuged for 20 min at 1,400 RCF. The cell band above the interface between the Histodenz and HBSS was harvested, washed in PBS, and plated in growth medium. One week after seeding, immune and epithelial cells were depleted by anti-EpCAM (Miltenyi, 130-105-958) and anti-CD45 (Miltenyi, 130-052-301) magnetic beads, and transferred to LS columns (Miltenyi, 130-042-401). For gene expression measurements, PSCs were then cultured for 3 days and mRNA was isolated.

**Organoid lines derived from primary pancreatic KPC tumors.** KPC organoids provided by the laboratory of David Tuveson[13] were cultured in Corning® Matrigel® Growth Factor Reduced (GFR) Basement Membrane Matrix, Phenol Red-free, LDEV-free, (Corning, 365231) with complete organoid medium[117]. Silencing of *Brca2* in KPC organoid lines was achieved by lentiviral infection as described above for the KPC cell-lines. Conditioned medium was collected following 3-4 days of culture with 5% FBS DMEM.

## PSC 3D cultures
For 3D culture, 4×10⁴ cells were seeded in Matrigel® GFR in growth medium for 4 days. Medium was changed to KPC-sh*Brca2* or KPC-shControl conditioned medium or to their own conditioned medium as control for 4 additional days and cells were either harvested for RT-PCR or RNA-seq. For HSF1 inhibition, 3 nM CMLD011866 (aglaroxin C)[78,79] was added every 2 days.

## In vivo tumor model
KPC orthotopic PDAC tumors were established as previously described[118]. Briefly, C57BL/6 J 8-week males were anaesthetized, and a small incision was made in the left part of the abdomen. Cancer cells (2×10⁴ KPC-sh*Brca2* or KPC-shControl per mouse) were suspended in Matrigel (Becton Dickinson), diluted 1:1 with cold PBS (total volume of 40 µL), and injected into the pancreas using a 26-gauge needle. The abdominal wall and then the skin were closed with Surgibond (Vetmarket, Israel). Buprenorphine was administered 30 min prior to the injection, and on the following day. Mice were sacrificed and tumors were collected for further analysis at 2-3 weeks postinjection. The maximum tumor volume of 2000 (mm)³ was not reached in any experiment.

## Sirius red assay
2×10⁴ PSCs were seeded in 96-wells and cultured overnight with growth medium. The medium was then replaced with conditioned medium from KPC-sh*Brca2* organoids, KPC-shControl organoids, or with control medium (DMEM 5% FBS, P/S) and the cells were incubated at 37 °C for 4 days. The medium was then aspirated and Sirius red/ fast green staining (Chondrex, cat. #9046) was performed according to the manufacturer instructions. Briefly, cells were washed with PBS and incubated in 100 µL Kale fixative for 10 min, after which the fixative was aspirated, the cells were washed with PBS and 100 µL of Sirius red/fast green dye was added for 30 min. Samples were imaged with an inverted Leica DMI8 wide-field (Leica Microsystems, Mannheim, Germany), Leica DFC7000GT monochromatic camera, 20x/0.8 Air. The

dye was then aspirated, the cells were washed, extraction buffer was added, and OD values at 540 nm and 605 nm were read with a spectrophotometer. The amount of collagen per sample was calculated using the following formula:

$$\frac{\text{OD540 value} - (\text{OD605 value}*0.291)}{0.0378}$$

## Immunohistochemistry of human tissues
4-µm FFPE sections from PDAC tumors were deparaffinized and treated with 1% $H_2O_2$. Antigen retrieval was performed using citrate buffer (pH 6.0) for all antibodies, except for MUC5B and HLA-DR (for which Tris-EDTA buffer (pH 9.0) was used). Slides were blocked with 10% normal horse serum (Vector Labs, S-2000) and the antibodies listed in Supplementary Data 12 were used. Visualization was achieved with 3,3'-diaminobenzidine as a chromogen (Vector Labs, SK4100). Counterstaining was performed with Mayer hematoxylin (Sigma Aldrich, MHS16). Images were taken with a Nikon Eclipse Ci microscope (Fig. 1a) or scanned by the Pannoramic SCAN II scanner, with 20×/0.8 objective (3DHISTECH, Budapest, Hungary) (Fig. 2b, c).

## Multiplexed Immunofluorescent (MxIF) staining and imaging of human tissues
**MxIF staining.** FFPE sections from 22 *BRCA*-WT and 14 *BRCA*-mut PDAC patients were deparaffinized, and fixed with 10% neutral buffered formalin. Antigen retrieval was performed using citrate buffer (pH 6.0) for all antibodies, except for MUC5B and HLA-DR (for which Tris-EDTA buffer (pH 9.0) was used). Slides were then blocked with 10% BSA + 0.05% Tween20 and the antibodies listed in Supplementary Data 11 were diluted in 2% BSA in 0.05% PBST and used in a multiplexed manner with OPAL reagents (AKOYA BIOSCIENCES). All primary antibodies were incubated overnight at 4 °C, except for αSMA which was incubated for 1.5 hrs at RT. Briefly, following primary antibody incubation, slides were washed with 0.05% PBST, incubated with secondary antibodies conjugated to HRP, washed again and incubated with OPAL reagents. Slides were then washed and antigen retrieval was performed as described above. Then, slides were washed with PBS and stained with the next primary antibody or with DAPI at the end of the cycle. Finally, slides were mounted using Immu-mount (#9990402, Thermo Scientific).

**MxIF imaging.** FFPE samples were imaged with a LeicaSP8 confocal laser-scanning microscope (Leica Microsystems, Mannheim, Germany), equipped with a pulsed white-light and 405 nm lasers using a HC PL APO ×40/1.3 oil-immersion objective and HyD SP GaAsP detectors. The following fluorophores and parameters were used: DAPI (Ex. 405 nm Em. 424–457 nm); Opal 520 (Ex. 494 nm Em. 510–525 nm); Opal 570 (Ex. 568 nm Em. 575–585 nm); Opal 620 (Ex. 588 nm Em. 601–616 nm); Opal 650 (Ex. 638 nm Em. 647–664 nm); Opal 690 (Ex. 670 nm Em. 725–794 nm); and pinhole of 1 AU. Samples were acquired with a pixel size of 0.142 µm/pixel.

**MxIF analysis.** Images were analyzed using Fiji image processing platform[119]. For all panels of MxIF staining 3-5 images were obtained per patient. For each image, five slices were Z projected (max intensity) and linear spectral unmixing was performed. We used two main methods for image analysis—object-based analysis and pixel-based analysis. Object-based analysis was applied for co-expression studies, such as in Supplementary Figure 1c–e, in which we aimed to determine whether our CAF markers mark different cells. Since these proteins might be expressed in the same cell, but not necessarily at the same pixel, we used object-based analysis, in which cells' borders are inferred based on the nuclei shape by DAPI staining. When assessing the abundance of each marker on its own, we used pixel-based analysis, as

some of these proteins are secreted and thus may be underrepresented if analyzed by cell structure inference rather than by analyzing the whole image. To assess the CAF composition (Fig. 1) each channel was thresholded to create a mask of its area. The area of each CAF marker within the CD45- CK- area was calculated by pixel-based analysis and divided by the total region of interest (ROI), as defined by the CD45- CK- area. For the assessment of CAF subtype ratios, values were logged and averaged per patient. To study the discrete expression of the different CAF markers (Supplementary Figure 1b–d) we performed an object-based analysis using the QuPath software[120]. Briefly, using a training image, each cell marker classifier was trained independently of all the other markers. CD45+ and CK+ cells were excluded. Then, the number of positive cells for each marker was calculated. The ratio of positive cells for each marker (αSMA, CLU, HLA-DR) was defined as N/A if there were less than 10 positive cells of that marker in that image. If there were more than 10 positive cells within the image, all 1st and 2nd order overlap ratios (relative to the chosen marker) were calculated. All images per patient were averaged. To analyze TF activation (Fig. 3) we used the Fiji image processing platform. First, we defined ROIs to exclude all cancer cells. Then, we detected nuclei of all stromal cells using the DAPI channel. Then, the mean intensity of each TF in all stromal cell nuclei per image was calculated. For each patient, the average intensity of all images was calculated. To analyze the expression of MUC5B and SERPINA1 within CLU+ and αSMA+ CAFs we used the Fiji image processing platform. First, we excluded all CK+ area. Then, the area of each of the CAF markers and secreted proteins was thresholded to create a mask of its area. Then, the area of CLU or αSMA out of MUC5B or SERPINA1 was calculated.

### Single-cell validation

Two human PDAC single-cell datasets[12,16] were analyzed using the Seurat (V4.0) R toolkit[56,121]. Pathway analysis was performed using Metascape[122]. UMAP images displaying gene expression were plotted using a minimum cutoff of the 10'th quantile.

Peng et al. dataset[16]—All cells that were defined as 'Fibroblast_cell' or 'Stellate_cell' in the original dataset were analyzed. Non-tumor samples and two samples with less than 50 Fibroblast or Stellate cells were filtered out. All other functions were run with default parameters. This yielded a large and comprehensive dataset of 11,010 cells from 22 patients. Harmony integration[123] with default parameters was used to minimize the patient batch effect, and shared nearest neighbor (SNN) modularity optimization-based clustering was then used with a resolution parameter of 0.14[124]. Two clusters were excluded from further analysis, one had less than 10 cells and the other is a cluster that expresses the pericyte marker, MCAM.

Elyada et al. dataset[12]—All cells that were originally defined as iCAF or myCAF were analyzed, yielding 972 cells from 10 patients. All other functions were run with default parameters. Harmony integration[123] with default parameters was used to minimize the patient batch effect, and shared nearest neighbor (SNN) modularity optimization-based clustering was then used with a resolution parameter of 0.1, which resulted in 4 clusters, following the exclusion of two clusters that had less than 10 cells, each.

### Pathway enrichment analysis

Pathway enrichment analysis was performed using Metascape[122] to analyze the DE genes in the LCM RNA-seq results, as well as the different clusters of the single-cell and bulk data.

### Pixel classification of H&E stained slides from PDAC patient samples

H&E slides were scanned by the Pannoramic SCAN II scanner, with 20×/0.8 objective (3DHISTECH, Budapest, Hungary). Quantification of CAF-rich, cancer-rich, and immune-rich regions within the tumor area of each section was done by QuPath (version 0.2.3)[120] using pixel classification. The classifier method used was Artificial neural network (ANN_MLP) with high resolution. The same classification parameters were used for all images.

### Laser capture microdissection of human PDAC samples

Fresh frozen blocks of *BRCA*-WT and *BRCA*-mut PDAC tumors were obtained from MSKCC (Supplementary Data 1). 7 mm sections were sliced in a cryostat and placed on PEN Membrane Glass Slides (Thermo Fisher Scientific, LCM0522). Then, sections were stained using the Histogene™ LCM Frozen Section Staining Kit (Thermo Fisher Scientific, KIT0401) and stromal regions were dissected. Immune islands, cancer cells and blood vessels were excluded from microdissection. Slides were left to dry for 5 min at RT followed by microdissection using the Arcturus (XT) laser microdissection instrument (Thermo Fisher Scientific, #010013097). Infrared capture was used to minimize RNA damage. CapSure Macro LCM caps (Thermo Fisher Scientific, #LCM0211) were used to capture microdissected tissue. Microdissected tissue from each sample was incubated for 1 h in 65 °C in the lysis buffer of the RNA extraction kit and frozen at −80 °C. RNA extraction was performed using the PicoPure™ RNA Isolation Kit (Thermo Fisher Scientific, KIT0204) according to the manufacturer's instructions.

### Library preparation and RNA-sequencing of LCM samples

Libraries were prepared using the SMARTer Stranded Total RNA-Seq v2-Pico Input Mammalian Kit (Takara Bio USA, #634415) according to the instructions of the manufacturer. Libraries were sequenced on Illumina NextSeq 500, at 25 M reads per sample.

### Differential expression analysis of LCM samples

The DeSeq2 package[125] was used to identify DE genes between *BRCA*-mut and *BRCA*-WT samples, and the FDRtool[126] was used to compute local FDR values from the p-values calculated using DeSeq2. Genes with a fold change (FC) >= 1.5 and false discovery rate (FDR) < 0.05 were considered significantly differentially expressed between groups. For comparison of sh*Brca2* and shControl samples, genes were significant at FC >= 0.5, FDR < 0.1 and basemean value > 5. Batch biases were corrected using Deseq2 package as RNA extraction and library preparation were performed in two batches.

### Isolation of CAFs from KPC tumors

C57BL/6 J mice were orthotopically injected with 2×10⁴ KPC-sh*Brca2* or KPC-shControl cells. At 2-3 weeks following injection animals were sacrificed, and tumors were excised, dissociated, minced and incubated with enzymatic digestion solution containing 2.5 mg/mL collagenase D (Sigma Aldrich, 11088866001), 70 U/mL Dnase, and 1 mM HEPES (Biological Industries) in RPMI 1640 (Biological Industries, 01-100-1 A) for 40 min at 37 °C. To enrich for stromal cells, single-cell suspensions were incubated with anti-EpCAM (Miltenyi, 130-105-958) and anti-CD45 (Miltenyi, 130-052-301) magnetic beads, transferred to LS columns (Miltenyi, 130-042-401) and the stromal enriched (CD45, EpCAM depleted) flow-through was collected and pelleted.

### Bulk RNA sequencing of CAFs, PSCs and KPC cells

**CAFs**. Two to three weeks following KPC shControl or KPC sh*Brca2* orthotropic injection into the pancreas the mice were sacrificed, PDAC tumors were excised, digested into single cells and suspended in MACS buffer. The cell suspension was depleted of CD45+ and EpCAM+ cells using manganic beads (Miltenyi, 130-105-958 and 130-052-301) and LS columns (Miltenyi, 130-042-401). The CAF-enriched flow through was then stained for FACS sorting with the following antibodies: CD45-FITC, CD31-FITC, EpCAM-FITC, PDPN-APC, and Ghost Dye-violet 450 for detection of live cells. To collect CAFs, the following gating strategy was applied: CD45⁻/CD31⁻/EpCAM⁻/PDPN⁺ (as PDPN

was previously shown to be a global CAF marker, including *Clu*[+] CAFs, in mice[12,14]. -10000 CAFs were sorted into 40 μL of lysis/binding buffer (Life Technologies) and mRNA was isolated using Dynabeads oligo (dT) (Life Technologies).

**PSCs.** PSCs were cultured in 3D as described above, and mRNA was extracted from the culture using Dynabeads® mRNA DIRECT™ Purification Kit (Thermo-Fisher scientific cat# 61012).

**KPC.** 200,000 KPC-sh*Brca2* or KPC-shControl cells were seeded in 6-wells and cultured in growth medium. After two days the cells were stained with propidium iodide (PI, 1:1000) and then 10,000 PI[-] cells were FACS sorted into 40 μL of lysis binding buffer (Thermo-Fisher scientific cat# 61012), and mRNA was isolated using Dynabeads oligo (dT).

RNA libraries of CAFs, PSCs and KPCs were prepared and sequenced on Illumina NextSeq 500, using the MARS-seq protocol, as previously described[90].

## Clustering and differential expression analysis for bulk RNA-seq of CAFs, PSCs, and KPCs

Raw read counts were processed and normalized utilizing the Deseq2 pipeline, using Likelihood ratio test for the paired comparisons. For CAF and KPC samples, genes were considered DE between sh*Brca2* and shControl and used for downstream analysis if their Padj < 0.1, absolute log2 FC > 0.5, and basemean value greater than 5. Then, pathway analysis was performed on the genes in each cluster using Metascape[122], and prediction of upstream regulators was performed using IPA[62].

For PSCs, two steps of differential expression and pathway analysis were performed: First, we extracted genes that were differentially expressed between any two of the three groups (sh*Brca2*-CM, shControl CM and Growth medium), and the resulting p-values were adjusted using FDR. Genes with any Padj < 0.1 and absolute log2 FC > 0.5 (for the same comparison), and a basemean value greater than 5 (*n* = 1320; Fig. 5a and Supplementary Data 8), were used for pathway analysis on each of the 4 clusters in Fig. 5a. Next, to extract genes upregulated only in sh*Brca2*-CM treated PSC, the DE genes in the PSC dataset (*n* = 1320) were filtered for Padj < 0.1 and log2FC < −0.5 for sh*Brca2 vs.* shControl, and for Padj <0.1 and log2FC < −0.5 for sh*Brca vs* growth medium. To extract genes upregulated only in shControl-CM treated PSC, the DE genes in the PSC dataset (*n* = 1320) were filtered for Padj < 0.1 and log2F C > 0.5 for shControl *vs* sh*Brca2*, and for Padj < 0.1 and log2F C > 0.5 for shControl *vs* growth medium (See Supplementary Data 8 tab 6).

## CIBERSORTx

To estimate the fraction of fibroblasts in the LCM-dissected samples, we used the computational deconvolution tool, CIBERSORTx, that estimates the relative abundance of individual cell types in a mixed cell population based on single cell RNA-seq profiles[57]. The single-cell human PDAC dataset by Peng et al.[16] was used as a reference to the cell type signatures. Then, we applied CIBERSORTx to estimate the distribution of immune cell populations within these samples using LM22, a validated leukocyte gene signature matrix[127]. The quantile normalization was disabled as recommended by the CIBERSORTx web interface (https://cibersortx.stanford.edu/). Permutations were set to 500.

## Proteomic analysis of human exosomes

Fresh pancreatic cancer tissue and peritumoral non-involved pancreas tissue were cut into small pieces and cultured for 24 h in serum-free RPMI, supplemented with penicillin (100 U/mL) and streptomycin (100 μg/mL). Conditioned medium was processed for exosome isolation. Exosomes were purified by sequential ultracentrifugation as previously described[61,128]. Briefly, cell contamination was removed

from resected tissue culture supernatant by centrifugation at 500x *g* for 10 min. To remove apoptotic bodies and large cell debris, the supernatants were then spun at 3000x *g* for 20 min, followed by centrifugation at 12,000x *g* for 20 min to remove large microvesicles. Finally, exosomes were collected by ultracentrifugation twice at 100,000x *g* for 70 min. Five micrograms of exosomal protein were used for mass spectrometry analysis[61]. High resolution/high mass accuracy nano-LC-MS/MS data was processed using Proteome Discoverer 1.4.1.14/Mascot 2.5. Human data was queried against the Uni-Prot's Complete HUMAN proteome.

## CM boiling experiment

CM was collected from KPC-shControl and KPC-sh*Brca2* organoid cultures following 4 days in culture. CM was boiled for 20 min and added to PSCs grown in 3D cultures for 72 h. *Clu* expression in PSCs was analyzed by RT-PCR.

## Mass spectrometry of organoid CM

Serum-depleted CM was collected from 8 KPC-shControl and 8 KPC-sh*Brca2* organoid cultures following 4 days in culture. Mass spectrometry was carried out at the De Botton Protein Profiling institute of the Nancy and Stephen Grand Israel National Center for Personalized Medicine, Weizmann Institute of Science. Samples were concentrated and denatured using urea and subjected to tryptic digestion. The resulting peptides were analyzed using nanoflow liquid chromatography (nanoAcquity) coupled to high resolution, high mass accuracy mass spectrometry (Q Executive HF). Each sample was analyzed on the instrument separately in a random order in discovery mode. Raw data was processed using MaxQuant version 2.0.1. Data was searched against the mouse protein sequences from UniprotKB. Quantification was based on the LFQ method, based on all peptides. The mass spectrometry proteomics data have been deposited to the ProteomeXchange Consortium via the PRIDE[129] partner repository with the dataset identifier PXD036629.

## Ingenuity pathway analysis

This tool generates multi-level regulatory networks by suggesting upstream regulators that may lead to the transcriptional changes evident in a dataset. To identify predicted upstream regulators of the *Brca2*-associated CAF transcriptional program we analyzed our DE gene dataset using the Causal Network tool in the Ingenuity Pathway Analysis (IPA) software[62].

## Real-time PCR

RNA was isolated using TRIzol reagent based on the TRI reagent user manual (Bio-Lab, 959758027100). Reverse transcription was done by High-Capacity cDNA reverse transcription kit (Cat 4368814, Thermo Fischer Scientific) according to the manufacturer instructions. Quantitative RT–PCR analysis was performed using Fast SYBR Green Master mix (Applied Biosystems, 4385610) or Taqman Fast Advanced Master Mix (Applied Biosystems, 4444556), and data was normalized to the house-keeping gene HPRT. The primer sequences for qPCR used in this study are provided in Supplementary Data 13.

## Aglaroxin C

((-)-Aglaroxin C (CMLD011866) was synthesized according to published protocols[78,79] and used at a concentration of 3 nM.

## Chromatin immunoprecipitation (ChIP) followed by qRT-PCR

Immortalized PSCs were treated with KPC-conditioned medium. 24 hrs later, PSCs were harvested for CHIP assay as described in[130]. Anti-HSF1 Ab (Cell Signaling, 4356 S) was used to immunoprecipitate HSF1, and normal rat IgG (Cell Signaling, 2729 S) was used as control. qRT-PCR was performed using the primers listed in Supplementary Data 13 to assess the binding of HSF1 to *Clu*, *Hsp90aa*, and *Hspa1a*. These

genomic primers were designed to flank an HSE site homologues to that reported to bind HSF1 in human cells[65]. (*Clu* HSE sequence: TTCCAGAAAGCTC, Mus musculus strain C57BL/6 J chromosome 14, GRCm39, 66205967- 66205980). Primers targeting an intergenic region (to which HSF1 is not expected to bind) were used as control.

## Conditioned medium for Chromatin IP

KPC cells were plated at a density of $15×10^4/cm^2$ in DMEM supplemented with 5% FBS and L-glu and Pen/Strep. 24 h later, the medium was replaced and cells were left to grow for an additional 48 h. The medium was then collected and filtered through 0.22 µm filters, and placed on top of PSC cultures.

## Analysis of HSF1 target genes

The HSF1base.org database of HSF1 targets[84] was queried for murine HSF1 target genes. For PSCs, this list was then matched with the filtered list of DE genes between sh*Brca2*-CM treated- and shControl-CM treated-PSCs from Supplementary Data 8, tab 6. For CAFs, the list of murine HSF1 targets was matched with DE genes from Supplementary Data 10 with p.adj < 0.05 and absolute log2FC >1.5. The statistical significance of the dependency between treatment (sh*Brca2*/ shControl) and HSF1 targets was tested using a Chi-square test.

## Ligand-receptor analysis

ICELLNET R package[87] was used to analyze normalized gene counts of CAFs derived from either shControl or sh*Brca2* tumors. Normalized gene counts that were above the median expression in each population were used. The ICELLNET ligand-receptor dataset of classifications was set to "Checkpoint" on the CD8+ T- cells that are provided by the package. The scores were calculated based on the expression of CAF ligands and CD8+ T-cell receptors[87].

## Flow cytometry for CAF ligands

KPC shControl and sh*Brca2* tumors were dissociated into single-cell suspensions and treated with red blood cell lysis buffer (Biolegend, 420301). Subsequently, cells were depleted of CD45 + and EpCAM+ cells as described above. For CAF enrichment, the CD45- and EpCAM-depleted fraction was incubated with PDPN–biotin antibody and the PDPN-enriched cell suspension was isolated with anti-biotin magnetic beads (Miltenyi, 130-090-485). Cells were stained for anti-CD45 and anti-EPCAM FITC (Miltenyi, 130-110-658, 130-117-752), anti-PDPN BV421 (Biolegend 127423), anti-PD-L1 APC (Biolegend 124312), anti-CD155 PE (Biolegend 132205), and anti-Nectin2 BV711 (BD Biosciences, 748049). Dead cells were excluded using Draq7 staining (Biolegend, 424001). FACS analysis was performed using flowjo software v.10.7.1 and CytExpert version 2.5.0.77

## T-cell activation assay

$2×10^4$ CAFs from either shControl or sh*Brca2* KPC tumors were plated in 96 wells in RPMI 1640 supplemented with 10% FBS. After 24 h, $2×10^4$ CD8 + T cells were isolated from normal spleens by a positive-selection kit (CD8a (Ly-2) Microbeads, mouse, Miltenyi 130-117-044), in the presence or absence of CD3/CD28 Dynabeads. For positive and negative controls, T-cells were activated without CAFs or cultured in the absence of CD3/CD28 beads. After 24 h of co-culture, magnetic beads were removed, and cells were analyzed by flow cytometry. CD25-BV711 and CD69-APC antibodies were used to determine CD8 + T-cell activation status, Ghost-Dye-Violet 450 (TONBO) was used to exclude dead cells. FACS analysis was performed using flowjo software v.10.7.1 and CytExpert version 2.5.0.77

## ECM structure analysis

**IF staining.** FFPE tumor sections from *BRCA*-WT and *BRCA*-mut PDAC patients ($n = 3$ each) were double stained with antibodies against αSMA (1:350) and CLU (1:100), using citrate buffer (pH = 6) for antigen retrieval. This was followed by the secondary antibodies AF488 anti mouse for αSMA and AF568 antirabbit for CLU. To detect nuclei staining, we incubated the slides for 10 min with DRAQ5 (ab108410). Slides were kept in PBS until imaging.

**Second harmonic generation (SHG) imaging.** FFPE slides were taken for SHG imaging using an upright Leica TCS SP8 MP microscope, equipped with external nondescanned detectors (NDD) HyD and acusto optical tunable filter (Leica microsystems CMS GmbH, Germany). The SHG signal was excited by a 885 nm laser line of a tunable femtosecond laser 680–1080 Coherent vision II (Coherent GmbH USA). The emission signal was collected using an external NDD HyD detector through a long pass filter of 440 nm. The transmitted signal was collected using a PMT detector in transmission position for general morphology. In addition, αSMA, CLU and DRAQ5 were imaged using excitation of Argon, DDPSS 561 and HeNe 633 lasers, with emission collection at 585-620 nm, 505-535 nm and 670–760 nm (respectively). Images were acquired using a format of 2048 × 2048 (XY) with an HC FLUOTAR L 25X/0.95 W VIS objective, and the following parameters: scan speed 600 Hz; Zoom 1; Line average- 4; bit depth- 16 FOV- X 442.86 µm, Y 442.86 µm; pixel size- 216 nm; Z step- 0.568 µm; Z stacks were acquired using the galvo stage, with 0.568 µm intervals. The acquired images were visualized using LASX software (Leica Application Suite XLeica microsystems CMS GmbH).

**SHG analysis.** For each image a ROI (100 µm X 100 µm) expressing high levels of either αSMA (in *BRCA*-WT) or CLU (in *BRCA*-mut) was chosen. ECM structures were analyzed according to a plug-in adapted from Wershof et al.[131]- TWOMBLI. Briefly, we started with Max projection (3 slices), and the desired ROI were cropped for further analysis. We ran the TWOMBLI with the following parameters: Contrast Saturation- 0.35; Min Line Width-10; Max Line Width- 15; Min Curvature Window- 30; Max Curvature Window- 50; Minimum Branch Length- 10; Maximum Display HDM- 50; Minimum Gap Diameter- 10.

## Statistical analysis

Statistical analysis and visualization were performed using R (Versions 3.6.0 and 4.0.0, R Foundation for Statistical Computing Vienna, Austria) and Prism 9.1.1 (Graphpad, USA). Statistical tests were performed as described in each Figure legend. Mann-Whitney test was used to analyze data that is not normally distributed. Student's t-test or ANOVA were used to analyze normally distributed data. Pearson's correlation coefficient was used to assess the association between two continuous variables. RNA-seq analysis of mouse cells was performed as described above. All other statistical tests were defined as significant if *p* value < 0.05 or FDR < 0.05 for multiple comparisons. "ns" in all Figures marks p-values greater than 0.05. Values that were more than 2 standard deviations of their group-mean were defined as outliers and excluded.

## Reporting summary

Further information on research design is available in the Nature Research Reporting Summary linked to this article.

## Data availability

Human PDAC single cell data analyzed in this paper can be found under GSA: CRA001160. Elyada et al. dataset is described in[12] (accession number phs001840.v1.p1). RNA-sequencing of LCM PDAC samples can be found in the dbGaP Authorized Access System (phs002994.v1); RNA-seq data from KPC, PSC and CAFs from KPC tumors in mice can be found under GSE200617. The mass spectrometry proteomics data have been deposited to the ProteomeXchange Consortium via the PRIDE[129] partner repository with the dataset identifier PXD036629. The remaining data are available within the Article, Supplementary Information or Source Data file. Source data are provided with this paper.

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

## Acknowledgements

Bioinformatic analyses were assisted by Ester Feldmesser, Ron Rotkopf, and Irit Orr (WIS). The authors thank Ela Elyada and Giulia Biffi for their guidance with the organoid system, and all members of the Scherz-Shouval lab for their valuable input. We thank Ofra Golani at the MICC cell observatory, WIS, and Dr. Liat Alyagor, Immunohistochemistry unit, WIS for their assistance with imaging. We thank Dr. Yishai Levin from the De Botton Protein Profiling institute of the Nancy and Stephen Grand Israel National Center for Personalized Medicine, WIS, for his assistance with mass spectrometry analysis. We thank Ms. Ariela Tomer for assistance in patient recruitment and sample collection. RSS is supported by the Thompson Family Foundation, ISF grant 395/21, ERC grant 754320, the Laura Gurwin Flug Family Fund, the Peter and Patricia Gruber Awards, the Comisaroff Family Trust, the Estate of Annice Anzelewitz, and the Estate of Mordecai M. Roshwal. RSS is the incumbent of the Ernst and Kaethe Ascher Career Development Chair in Life Sciences. LS was supported by the Rising Tide Foundation. DK was supported by MSK Center Core Grant P30 CA008748 and data science grant. Work at Boston University was supported by NIH grants R35GM118173 and U01TR002625.

## Author contributions

L.S., A.B.S., and M.P.F. designed, performed and analyzed experiments and wrote the manuscript. G.F., O.L.G., D.B. and C.L. designed and performed experiments. S.N., Y.S. and S.M. designed and performed bioinformatic and statistical analysis. R.N. assisted with image acquisition and designed image analysis. L.E.B., W.Z. and J.A.P. provided reagents and intellectual input, and J.A.P. secured funding. H.S.K., L.B. and N.L. performed experiments and provided intellectual input. R.S., H.B. and R.H. performed experiments. W.R.J., E.L.L., T.G. and C.A.I.D. provided clinical samples and intellectual input. N.S. directed and designed bioinformatic analysis. D.L. directed and designed experiments, provided intellectual input and wrote the manuscript. D.A.T. provided reagents and intellectual input. D.K. provided clinical samples and intellectual input, secured funding and wrote the manuscript. R.S.S. designed and directed the study, designed and analyzed experiments, secured funding and wrote the manuscript.

## Competing interests

The authors declare no competing interests.

## Additional information

Lee Shaashua [1,13], Aviad Ben-Shmuel [1,13], Meirav Pevsner-Fischer [1,13], Gil Friedman[1], Oshrat Levi-Galibov[1], Subhiksha Nandakumar[2], Debra Barki[1], Reinat Nevo[1], Lauren E. Brown[3], Wenhan Zhang[3], Yaniv Stein[1], Chen Lior[1], Han Sang Kim[4,5], Linda Bojmar[4,6], William R. Jarnagin[7], Nicolas Lecomte[8], Shimrit Mayer[1], Roni Stok[1], Hend Bishara[1], Rawand Hamodi[1], Ephrat Levy-Lahad[9], Talia Golan[10], John A. Porco Jr.[3], Christine A. Iacobuzio-Donahue[8], Nikolaus Schultz[2], David A. Tuveson[11], David Lyden[4], David Kelsen[12] & Ruth Scherz-Shouval[1]✉

[1]Department of Biomolecular Sciences, The Weizmann Institute of Science, Rehovot, Israel. [2]Human Oncology and Pathogenesis Program, Memorial Sloan Kettering Cancer Center, New York, NY, USA. [3]Department of Chemistry and Center for Molecular Discovery (BU-CMD), Boston University, Boston, MA, USA. [4]Children's Cancer and Blood Foundation Laboratories, Departments of Pediatrics, and Cell and Developmental Biology, Drukier Institute for Children's Health, Meyer Cancer Center, Weill Cornell Medicine, New York, NY, USA. [5]Yonsei Cancer Center, Division of Medical Oncology, Department of Internal Medicine, Graduate School of Medical Science, Brain Korea 21 Project, Severance Biomedical Science Institute, Yonsei University College of Medicine, Seoul, Korea. [6]Department of Biomedical and Clinical Sciences, Linköping University, Linköping, Sweden. [7]Hepatopancreatobiliary Service, Department of Surgery, Memorial Sloan Kettering Cancer Center, New York, NY, USA. [8]David M. Rubenstein Center for Pancreatic Cancer Research, Memorial Sloan Kettering Cancer Center, New York, NY, USA. [9]The Fuld Family Medical Genetics Institute, Shaare Zedek Medical Center, Jerusalem, Faculty of Medicine, The Hebrew University of Jerusalem, Jerusalem, Israel. [10]Oncology Institute, Sheba Medical Center at Tel-Hashomer, Tel Aviv University, Tel Aviv, Israel. [11]Cancer Center, Cold Spring Harbor Laboratory, Cold Spring Harbor, New York, NY, USA. [12]Gastrointestinal Oncology Service, Memorial Sloan Kettering Cancer Center, Weill Cornell Medical College, New York, NY, USA. [13]These authors contributed equally: Lee Shaashua, Aviad Ben-Shmuel, Meirav Pevsner-Fischer. ✉e-mail: ruth.shouval@weizmann.ac.il

