## [Peer Review File · Nature Communications]

Reviewers' Comments:

Reviewer #1:

Remarks to the Author:

In this manuscript by Shaashua, Pevsner-Fischer and colleagues, the authors investigate the relationship between BRCA status and CAF transcriptional state. This is a timely study and one likely to be of great interest to the pancreatic cancer and tumor microenvironment communities. As the authors note, a substantial population of pancreatic cancer patients harbors BRCA mutations, supporting the significance of this work. From a broader perspective, this study provides compelling evidence that tumor mutational status plays a causal role in shaping the pancreatic tumor microenvironment with respect to CAF phenotype, in agreement with a small number of prior studies but with far-reaching implications (and this is entirely novel with respect to BRCA status in pancreatic cancer). To investigate the impact of BRCA status on the PDAC stroma, the authors used a diverse array of robust approaches including LCM followed by RNA-seq and multiplex IHC on patient specimens of known BRCA status, mouse models of pancreatic cancer, and cell culture experiments exposing CAF precursors to secreted factors from cancer cells expressing or lacking BRCA2. Together, these analyses convincingly implicate loss of BRCA in transcriptional rewiring of pancreatic CAFs, including induction of Clusterin and additional alterations involved in diverse CAF functions. Further, the authors convincingly link BRCA mutation to stromal activation of HSF1, an unexpected and exciting discovery. The data are of very high quality throughout the manuscript, diverse model systems are employed to support the authors' claims, statistics are appropriate throughout, and the conclusions are generally supported by the data. A limited number of additional experiments or analyses would help to strengthen the central claims and clarify these findings, particularly with respect to some noted differences across species and the nature of the transcriptional shift in CAFs in the setting of BRCA mutation.

Specific comments:

- 1) While the distinction between α SMA⁺ CAFs and Clu⁺ CAFs is very clear from tissue staining and scRNA-seq analyses, the link between Clu expression and immune-modulatory or inflammatory gene expression is less clear. This certainly does not detract from the striking impact of BRCA mutation on CAF transcriptional programs, but some clarification in the text and some additional validation would be meaningful to aid interpretation with respect to whether Clu induction and additional transcriptional alterations indeed coincide with expansion of iCAF per se in BRCA mutant tumors. The authors clearly state that Clu associates with iCAFs among PDAC patients but with apCAFs in mouse models—is there another established iCAF marker with an antibody suitable for IHC that can be used to validate Clu as an iCAF marker in patient specimens at the protein level and that is also elevated in BRCA mutant tumors? Alternatively, does Clu associate with genes in the cell adhesion and MAPK pathway categories identified in mouse models and cell culture assays in Figure 5 upon BRCA perturbation? This added clarification may help make the study more cohesive, or may further highlight meaningful differences across species for the field to keep in mind.
- 2) The MUC5B and SERPINA1 data in Figure 2 are very nice. Are these iCAF markers? Or enriched in BRCA mutant stroma but not necessarily part of the iCAF signature? Either way this would be informative to state in the manuscript text.
- 3) To strengthen the results in Figure 4C-F, are other genes besides Clu that are differentially expressed in BRCA mutant versus control stroma differentially expressed in Hsf1-null PSCs compared to wild-type?
- 4) As Clu-expressing CAFs are pro-angiogenic in breast cancer stroma (PMID: 30470719), is the abundance of CD31⁺ blood vessels increased in BRCA mutant tumors?
- 5) Is more nuclear HSF1 observed in murine shBrca2 tumors compared to control tumors? This would be a meaningful complement to patient data in Figure 3A-C and help support a causal role for cancer cell-intrinsic BRCA status in stromal activation of HSF1.
- 6) Minor: In panel 6D, "CM" should be removed from the x-axis labels.

Reviewer #2:

Remarks to the Author:

In the manuscript by Shaashua et al., the authors investigate the effect of BRCA1/2 mutations in CAF populations. The authors compared CAFs in tumors from BRCA wild-type and mutant patients and show an increase in CLU+ positive CAFs compared with other CAF subtypes in BRCA-mutant patients. Based on the assumption that CLU is upregulated on the transcriptional level in BRCA-mutant tumors, the authors implicate the stress-responsive TF HSF1 in upregulation of CLU. The data to support transcriptional upregulation of CLU in BRCA-mutant tumors is thin, while the evidence for wholesale “transcriptional re-wiring” is nonexistent. Beyond that, the functional upshot of any change in CAFs that does occur in BRCA-mutant tumors is not explored.

Major Points

1. The authors suggest that BRCA-mutant tumors are enriched for CLU-positive CAFs, though their RNA-seq data in microdissected stromal samples seems to undermine this. First, CLU itself is not significantly upregulated in the stroma of BRCA-mutant tumors. Even if we acknowledge a trend towards higher CLU, the RNA-seq gives us little reason to believe that a wholesale shift in CAF subtypes is occurring in BRCA-mutant tumors. Only 40 genes are significantly altered between BRCA wild-type and mutant tumors. A cursory comparison of gene expression in CAF populations from Peng et al. (provided by the authors) with the altered genes in their RNA-seq seems to yield no alignment with a particular subtype of CAF. The authors should use GSEA to compare their RNA-seq with gene expression in the single cell data. Similarly, the altered genes in PSCs induced by Brca depletion in cancer cells bears almost no resemblance to stromal changes in BRCA-mutant tumors, again GSEA could show a correspondence. If neither comparison yields a similarity, we are left wondering about the relevance of the single cell and in vitro RNA-seq data the authors include.

2. The authors provide IHC for MUCB and SERPINA1 and tumor-derived exosome data to “validate” their RNA-seq findings. This is irrelevant, as the question is not whether the proteins are expressed, but rather whether they are increased in BRCA mutant vs wild-type stroma. Indeed they are not, as stated in lines 341-342. The authors then resort to looking at MUC5B levels only in CLU+ CAFs, to finally find a difference. The RNA-seq data was obtained using whole stroma, so why should this validation require only looking at CLU+ CAFs, especially if CLU+ CAFs are so greatly increased in BRCA-mutant tumors?

3. The authors fail to provide any evidence that the supposed changes in CAFs have any functional importance. The tumor growth changes shown in Figure 6D could as easily be due to tumor cell intrinsic reasons as to CAF-mediated changes.

Minor point

1. It is unclear whether the data shown in Figure 2F is from multiple patients or multiple fields of view. How many patients were compared?

Reviewer #3:

Remarks to the Author:

The manuscript by Shaashua, Pevsner-Fischer and colleagues describes differences in the stromal cell landscape of pancreatic cancer patients with and without BRCA1/2 mutations. Using H&E, IHC, MxIF and single-cell RNA-seq the authors identify several types of cancer-associated fibroblasts, of which a subset expressing Clusterin (CLU) is enriched in BRCA-mutated tumors. Through a series of in vitro experiments, the authors provide evidence that this phenotype can be induced by transcription factor HSF1 in CAFs and leads to a switch from a myofibroblastic to an immune-regulatory phenotype of CAFs – mediated by BRCA-mutant tumor cell to fibroblast crosstalk.

Overall, this manuscript is a great example how the TME is at least in part shaped by tumor intrinsic features. It reveals that BRCA1/2 mutations (found in up to 8% of patients with sporadic pancreatic cancer and in much higher frequencies in breast and ovarian cancer) can directly modulate the TME, as shown through CAF phenotypes. As such, it represents a significant advancement for the field.

While I think the work is overall a strong mechanistic study, I would like to encourage the authors to take some of the points listed below into account before resubmitting the revised version of their article.

Major:

1. The authors analyze single-cell data from Peng et al. and find three major CAF subtypes. One of these is characterized by high levels of MHC II genes and is therefore considered antigen presenting (ap)CAFs. Looking into Suppl. Table 2, I would interpret this cluster (cluster 3) as immune cell doublets. These cells express considerable amounts of CD45, CD3, CSF1R, CD14. Additionally, they also express MCAM (which is confusing because the main text says MCAM+ cells were excluded) and RGS5, so they might in fact be doublets of immune cells and pericytes. [Tools like scGate (PMID 35258562) to reliably "gate" single cell data may help to clean this up]. I am not convinced that the cluster the authors suggest as apCAFs is not a technical artifact.

2. Given the point above, and the manuscript by some of the authors: Elyada et al (PMID 31197017) stating that ~20% of CAFs with detectable MHC II expression do not form a separate cluster, I am a little concerned about the distinct stains for HLA-DR, SMA and CLU presented in figure 1 and Suppl. Figure 1. These results seem contradictory to the results in Elyada et al to me? For the stains, I am missing a positive control that marks all fibroblasts (eg PDGFRA) - the authors aim to exclude non fibroblast cells (CD45- Cytokeratin-). In the case of SMA for example we know that pericytes are a main source of SMA expression so this strategy has shortcomings (that may extend beyond this example).

3. I am concerned about the conclusion the authors draw from Fig 4H that "Treatment with aglaroxin C abolished the induction of Muc5b expression". The comparison between shBrca2 + vs - aglaroxin is not significant and 2 of the 5 replicates in the + group show a comparable or higher Muc5b level to what is observed as the median in the - group. That is actually along the lines of their other experiment: "Additionally, Muc5B showed somewhat reduced expression in Hsf1 null mice but this result was not significant". Did the authors also perform a targeted immunoprecipitation PCR experiment for Muc5b promoter binding by HSF1 similar to what is shown for Clu (Fig. 4L)?

4. I am wondering if the authors can provide any insights into what drives differential HSF1 activation in fibroblasts between BRCA1/2 WT and mutant tumors? Do the differentially expressed genes in tumor cells from their mouse model (KPC-shBrca2 and KPC469 shControl cells) give any insights what might be secreted by one but not the other? Have the authors considered to mass spec the conditioned media?

Minor:

1. The authors excluded the normal adjacent tissue samples from their single-cell analysis. How is CLU expressed in normal adjacent tissue? Are the CLU high and low clusters from the same patients? Or are there between patient differences (e.g. some only have high, others only have CLU low CAFs)?

2. I don't understand how the top 20 genes for figure 5E were chosen. When I sort suppl Table 9 tab 1 by adjusted p-value, I see a different set of genes. Can the authors also comment on the purity of their sort (CSF1R and KRT7/8/18 come up as quite highly and differentially expressed and upregulated in shBRCA2). Maybe the authors could plot the expression of a few general fibroblasts markers (Pdgfra, Pdpn, ..) to show they are similar between the groups.

3. "We found that the expression of Clu was significantly lower in Hsf1 null PSCs compared to WT PSCs, while the expression of other CAF markers, such as Acta2 and Il6, was not altered (Figure 4E-F)." - How does this relate to expression correlation in their human laser capture microdissected CAF-rich region RNA-seq data? Is there a correlation between HSF1 and CLU expression levels?

4. Fig. 6D: Maybe I misunderstood, but I assume cells were orthotopically implanted into C57BL/6J mice. In that scenario the x axis label should not refer to CM?

Point-by-point response to reviewers' comments

Reviewer #1 (Remarks to the Author): with expertise in PDAC, CAF

In this manuscript by Shaashua, Pevsner-Fischer and colleagues, the authors investigate the relationship between BRCA status and CAF transcriptional state. This is a timely study and one likely to be of great interest to the pancreatic cancer and tumor microenvironment communities. As the authors note, a substantial population of pancreatic cancer patients harbors BRCA mutations, supporting the significance of this work. From a broader perspective, this study provides compelling evidence that tumor mutational status plays a causal role in shaping the pancreatic tumor microenvironment with respect to CAF phenotype, in agreement with a small number of prior studies but with far-reaching implications (and this is entirely novel with respect to BRCA status in pancreatic cancer). To investigate the impact of BRCA status on the PDAC stroma, the authors used a diverse array of robust approaches including LCM followed by RNA-seq and multiplex IHC on patient specimens of known BRCA status, mouse models of pancreatic cancer, and cell culture experiments exposing CAF precursors to secreted factors from cancer cells expressing or lacking BRCA2. Together, these analyses convincingly implicate loss of BRCA in transcriptional rewiring of pancreatic CAFs, including induction of Clusterin and additional alterations involved in diverse CAF functions. Further, the authors convincingly link BRCA mutation to stromal activation of HSF1, an unexpected and exciting discovery. The data are of very high quality throughout the manuscript, diverse model systems are employed to support the authors' claims, statistics are appropriate throughout, and the conclusions are generally supported by the data. A limited number of additional experiments or analyses would help to strengthen the central claims and clarify these findings, particularly with respect to some noted differences across species and the nature of the transcriptional shift in CAFs in the setting of BRCA mutation.

Response: We thank the reviewer for the positive assessment of our results and helpful suggestions.

Specific comments:

1) While the distinction between aSMA+ CAFs and Clu+ CAFs is very clear from tissue staining and scRNA-seq analyses, the link between Clu expression and immune-modulatory or inflammatory gene expression is less clear. This certainly does not detract from the striking impact of BRCA mutation on CAF transcriptional programs, but some clarification in the text and some additional validation would be meaningful to aid interpretation with respect to whether Clu induction and additional transcriptional alterations indeed coincide with expansion of iCAF per se in BRCA mutant tumors. The authors clearly state that Clu associates with iCAF among PDAC patients but with apCAF in mouse models—is there another established iCAF marker with an antibody suitable for IHC that can be used to validate Clu as an iCAF marker in patient specimens at the protein level and that is also elevated in BRCA mutant tumors? Alternatively, does Clu associate with genes in the cell adhesion and MAPK pathway categories identified in mouse models and cell culture assays in Figure 5 upon BRCA perturbation? This added clarification may help make the study more cohesive, or may further highlight meaningful differences across species for the field to keep in mind.

Response: Thank you for this constructive comment. We have now added a comprehensive functional characterization of the CLU⁺ CAF subtype that suggests an immune-regulatory role of this subtype, and a less myofibroblastic phenotype:

- To assess immune-modulation, we isolated CD8⁺ T-cells from spleens of naïve C57BL/6 mice and activated them in the presence of CAFs isolated from WT or *Brca2* deficient KPC tumors. We found that CAFs from KPC-sh*Brca2* tumors significantly repressed the activation of CD8⁺ T-cells relative to CAFs from KPC-shControl tumors as measured by CD69 and CD25 surface expression (Figure 6b-c).
- In the original manuscript we have shown that PSCs upregulate PD-L1 in response to KPC-sh*Brca2* CM. We extended our search for immune-modulatory cell surface proteins by analyzing our RNA-seq data from mouse CAFs using the ICELLNET receptor-ligand analysis tool (Noel et al, 2021). We found that *Pvr* (CD155), the ligand for the inhibitory checkpoint receptor TIGIT, was upregulated in CAFs derived from KPC-sh*Brca2* tumors vs KPC-shControl tumors (Figure 6d). To test whether this leads to differential cell-surface expression we performed FACS staining for the TIGIT ligands CD155 and Nectin2, as well as PD-L1, on CAFs freshly derived from KPC-sh*Brca2* vs KPC-shControl tumors. CAFs from KPC-sh*Brca2* tumors exhibited significantly higher cell-surface expression levels of CD155 and Nectin2 (as compared to KPC-shControl), and a similar trend was observed for the expression of PD-L1 (Figure 6e-g). Of note, Nectin2 (PVRL2) was also one of the upregulated immune regulatory genes in subcluster 2 (CLU^{high}) of the human Peng PDAC data set, further supporting the link between *Clu* and immune regulatory functions in mouse and human CAFs.
- Next, to further explore the changes in ECM deposition (that we have shown *in vitro* in the original manuscript) we assessed ECM organization in the vicinity of CLU⁺ or α SMA⁺ CAFs by performing second harmonic generation imaging in our patient cohort (Figure 6j-m). We found that the ECM associated with CLU⁺ CAFs is significantly different from that found in the vicinity of α SMA⁺ CAFs. α SMA-associated ECM was stiff, while CLU-associated ECM was “curly”, as measured by decreased perpendicular alignment, and increased curvature and branching indices (Figure 6k-m).

Overall, these findings support the definition of this subtype as iCAFs. These new analyses are included in Figure 6 and rows 551-596 of the manuscript:

Shaashua et al; Figure 6

Figure 6. *Brca2*-deficient cancer cells shift CAF functions. (a) Immortalized PSCs were seeded in Matrigel for 4 days in 10% FBS in DMEM. Conditioned media (CM) from PSCs or from KPC cells in which *Brca2* was silenced by shRNA or non-targeting shControl (KPC) was then added for an additional 4 days. 3nM of the HSF1 inhibitor, CMLD011866 (aglaroxin C), or PBS control was added to the conditioned media. *Pd11* were measured by qRT-PCR. Data are presented as mean \pm SEM. (b-c) CD8⁺ T-cells were isolated from spleens of naïve C57BL/6 mice, cocultured, and activated with anti-CD3/CD28 beads for 24 hours in the presence of CAFs isolated from either KPC-shControl or KPC-sh*Brca2* tumors. Subsequently, T-cells were subjected to FACS analysis for surface expression of CD69 and CD25. Data quantified are presented as mean \pm SEM of 6 KPC-shControl and 6 KPC-sh*Brca2* mice. (d) Scores of ligand-receptor binding were calculated using the ICELLNET R package (see Methods) to predict potential differential interactions between ligands of CAFs derived from shControl vs sh*Brca2* tumors with immune checkpoint receptors on CD8⁺ T-cells. (e-g) KPC-shControl and sh*Brca2* tumors were dissociated into single-cell suspensions. Dead cells were excluded using Draq7 staining. CD45⁺EpCAM⁺PDPN⁺ cells were isolated and stained with anti-CD155 (e), anti-Nectin2 (f), and anti-PD-L1 (g). Statistical significance was assessed by t-test. (h-i) PSCs were treated with CM derived from KPC-sh*Brca2* or KPC-shControl organoids, or with growth medium (DMEM with 5% FBS) as control for 4 days. Then, cultures were stained with Sirius red (SR) to assess collagen deposition. Collagen levels were calculated by measuring the absorbance at 540nm and 605nm wavelength (see Methods). Each point represents the average of sh*Brca2* or shControl normalized to the average of the growth medium control in each experiment. n=3 independent experiments, each representing an average of 5 technical replicates. t-test was performed on normalized values. (i) Representative images of PSCs stained with SR following 4 days treatment with CM derived from KPC-sh*Brca2* or KPC-shControl organoids. Scale bar=300 μ m. (j-m) FFPE tumor sections from *BRCA*-mut and *BRCA*-WT PDAC patients were stained by double staining for α SMA and CLU and imaged using Second harmonic generation (SHG) imaging. (j). Representative images are shown. DRAQ5 was used to stain nuclei. Scale bar- 100 μ m, or 25 μ m (inset). (k-m) Quantification of matrix pattern using the TWOMBLI plug-in (see Methods). The following parameters were analyzed: (k) Alignment - the extent to which fibers within the field of view are oriented in a similar direction; (l) Curvature- the mean change in angle moving incrementally along 40 μ m mask fibers; and (m) Branchpoint - the number of intersections of mask fibers in the image. (n) Schematic representation of the proposed model. Secreted factors from *BRCA*-mutated cancer cells induce *HSF1* activation in a subset of adjacent PSCs leading to their transcriptional rewiring into immune-regulatory CLU⁺ CAFs.

“The shift in CLU⁺/SMA⁺ CAF ratios in human patients, and the shift in Clu vs Acta2 expression in PSCs conditioned by Brca2-deficient cancer cells, suggest that CAFs of BRCA-mutated tumors may undergo a shift from myofibroblastic to immune-regulatory functions. Supporting this notion, Pd11, whose upregulation in PDAC CAFs was suggested to mediate T-cell immune suppression^{19, 84, 85}, was induced in PSCs by KPC-shBrca2 CM, but not by KPC-shControl CM (Figure 6a). Similar to Clu, the induction of Pd11 was inhibited by aglaroxin C, suggesting an increased immune-regulatory function for PSCs induced by BRCA-deficient cancer cells, in an HSF1-dependent manner (Figure 6a). To functionally test the ability of CAFs isolated from Brca2-deficient tumors to regulate immune cell function, we isolated CD8⁺ T-cells from spleens of naïve C57BL/6 mice, and activated them in the presence of CAFs from KPC-shControl or KPC-shBrca2 tumors (Figure 6b-c and Supplementary Figure 6a and b). T-cells activated in the presence of CAFs from Brca2-deficient tumors were significantly more repressed in their ability to upregulate the activation markers CD25 and CD69 compared to T-cells activated in the presence of CAFs from shControl tumors.

To search for potential factors that could be mediating the inhibitory effect of CAFs from shBrca2 tumors on T-cells, we mined our tumor-derived CAF RNA-seq data from KPC tumors using the ICELLNET receptor-ligand analysis tool employing a CD8⁺ T-cell-receptor dataset⁸⁶. ICELLNET is a computational tool that calculates a communication score for ligand-receptor interactions based on transcriptomic data and a database of potential ligand-receptor pairs. We applied ICELLNET to screen for immune modulatory surface ligands in CAFs that may inhibit T-cell activity, and found that CAFs from KPC-shBrca2 tumors scored higher than KPC-shControl CAFs

in the checkpoint signaling axis involving the TIGIT and CD96 T-cell inhibitory receptors, and their cognate ligands CD155 and Nectin1-3 (Figure 6d). To experimentally validate the cell-surface expression of inhibitory checkpoint markers on CAFs, we isolated CAFs from KPC-shControl or KPC-shBrca2 tumors, and performed FACS analysis using antibodies for PD-L1, CD155, and Nectin2. CAFs from KPC-shBrca2 tumors demonstrated higher cell surface expression of CD155 and Nectin2 (as compared to KPC-shControl), and a similar trend was observed for the expression of PD-L1 (Fig. 6e-g and Supplementary Fig. 6c), confirming the ICCELLNET receptor-ligand results and further supporting their role in suppressing T-cell function.

Next, we assessed myofibroblastic functions. To that end, we measured the ability of PSCs to secrete collagen, in-vitro, using Sirius Red staining. We found that PSCs conditioned by KPC-shBrca2 medium secreted significantly less collagen than PSCs conditioned by KPC-shControl medium (Figure 6h-i). To test the relevance of these findings in patients we assessed ECM organization in the vicinity of CLU⁺ or α SMA⁺ CAFs by second harmonic generation (SHG) imaging combined with MxIF staining in BRCA-WT vs BRCA-mut patients (Figure 6j-m). CLU-rich stromal regions that are abundant in BRCA-mut tumors demonstrated an altered ECM architecture, characterized by significantly reduced parallel alignment, and increased curvature and branching of collagen streaks compared to α SMA-rich regions (Figure 6j-m). Overall, our findings suggest that BRCA-mut cancer cells promote a stressful TME that leads to the activation of HSF1 in a subset of PSCs. These PSCs are reprogrammed into immune regulatory CLU⁺ CAFs, resulting in a different stromal landscape in BRCA-mut compared to BRCA-WT PDAC tumors (Figure 6n).”

2) The MUC5B and SERPINA1 data in Figure 2 are very nice. Are these iCAF markers? Or enriched in BRCA mutant stroma but not necessarily part of the iCAF signature? Either way this would be informative to state in the manuscript text.

Response: We thank the reviewer for this comment. Neither MUC5B nor SERPINA1 are known markers of fibroblast subtypes. While no interaction is known for CLU and MUC5B, an interaction between SERPINA1 and CLU was reported in two lung cancer cell lines (Ercetin et al., 2019). We added this information to the text (rows 286-289). Given that no direct interactions were found in our systems between CLU, MUC5B and SERPINA1, we assume that these findings are not directly related.

“SERPINA1 levels were elevated in PanIN3 lesions ⁶⁰ and correlated to CLU expression in two lung cancer cell lines ⁶¹, and MUC5B was identified in pancreatic main duct fluid collected at the time of surgical resection ⁶² but no known association with CLU was reported”.

3) To strengthen the results in Figure 4C-F, are other genes besides Clu that are differentially expressed in BRCA mutant versus control stroma differentially expressed in Hsf1-null PSCs compared to wild-type?

Response: We are happy to clarify this point. Clusterin is a chaperone, and as such it is regulated by HSF1 even in quiescent PSCs. We do not necessarily expect to see strong activation of non-

chaperone, CAF-related HSF1 targets under these conditions. We added a sentence clarifying this point to the manuscript (rows 379-380).

“CLU is an extracellular chaperone transcriptionally regulated by HSF1 in various contexts^{66, 72, 73} and upregulated in response to DNA damage^{74, 75}.”

We addressed the dependency on HSF1 of the transcriptional shift in PSCs exposed to KPC-shControl or KPC-shBRCA2 conditioned media (CM) by inspecting a publicly available dataset of HSF1 target genes (<https://hsf1base.org/>; Fig. 5e). We found a multitude of genes from this database (in addition to clusterin) that were enriched in KPC-shBrca2-treated PSCs relative to KPC-shControl.

In a separate effort to strengthen the link between BRCA-deficient cancer cells and activation of HSF1 and CLU, we performed mass-spec analysis of CM from KPC-shBrca2 vs KPC-shControl organoids (Fig. 4n-o). Six of the ten most differentially secreted proteins from KPC-shBrca2 vs KPC-shControl organoids were previously shown by us to be upregulated during colon inflammation in an HSF1-dependent manner (REG3G, REG3B, GC, SERPINH1, FN1, and PXDN; Levi-Galibov et al, 2020). We also found CLU itself in this list.

Taken together, these analyses strengthen the link between BRCA-deficiency and HSF1 activation.

This data was added to the manuscript in Figure 4n-o, and to the text (rows 422-440) as follows:

“In search for factors that may mediate this effect, we next analyzed the medium conditioned by KPC-shBrca2 cells. First, to test whether HSF1 and Clu upregulation is mediated by secretion of proteins, we boiled CM from KPC-shControl and KPC-shBrca2 organoids and treated PSCs with either un-boiled or boiled CM. Boiling of KPC-shBrca2 CM significantly reduced expression of Clu by PSCs (Fig. 4n), suggesting that this effect is mediated by a secreted protein(s). Next, we performed mass-spectrometry analysis of the organoid CM. The two most differentially secreted proteins from KPC-shBrca2 relative to KPC-shControl organoids were the Regenerating islet-derived (Reg) proteins, REG3B/G (Fig. 4o, Supplementary Table 7). REG3B/G are C-type secreted lectins that play active roles in pancreatitis and in the transition from pancreatitis to pancreatic cancer through different mechanisms, including induction of STAT3, RAF-MEK-ERK signaling, and immune cell modulation^{81, 82, 83, 84}. Of note, we have previously demonstrated an association between REG3B/G and HSF1 signaling, by showing that REG3B/G are upregulated during inflammation in the colon in an HSF1-dependent manner⁵⁴. In fact, six of the ten most differentially secreted proteins from KPC-shBrca2 vs KPC-shCont organoids were previously shown by us to be upregulated during colon inflammation in an HSF1-dependent manner (REG3G, REG3B, GC, SERPINH1, FN1, and PXDN)⁵⁴. We also found CLU itself in this list. These findings suggest that loss of BRCA2 in cancer cells leads to differential secretion of proteins resulting in activation of HSF1 in stromal fibroblasts, and, potentially, also in the cancer cells themselves.”

Figure 4. (n) PSCs were cultured in the presence of boiled or unboiled CM from KPC-shControl and KPC-sh*Brca2* organoids as described in (g-j). Expression levels of *Clu* in PSCs were subsequently measured by qRT-PCR. (o) Top 10 differentially expressed proteins (fold change sh*Brca2*/shControl) from CM of KPC-shControl and KPC-sh*Brca2* organoids as measured by mass-spectrometry.

4) As *Clu*-expressing CAFs are pro-angiogenic in breast cancer stroma (PMID: 30470719), is the abundance of CD31+ blood vessels increased in BRCA mutant tumors?

Response: Thank you for this comment. Our data indeed imply that there is an increased pro-angiogenic profile in CAFs isolated from KPC-sh*Brca2* tumors compared to KPC-shControl tumors, as “vasculature development” was among the enriched pathways in KPC-sh*Brca2* CAFs (Fig. 5d). To directly address this comment, we performed CD31 staining of *BRCA*-WT vs *BRCA2*-mut patient samples, and performed image analysis to quantify CD31+ cells, however we did not find significant differences between *BRCA*-WT and *BRCA*-mut tumors.

Additionally, we compared the enrichment of “Hallmark Angiogenesis Pathway” using GSEA in the *BRCA*-mut and *BRCA*-WT LCM-captured stroma. We found a trend of elevated enrichment in the *BRCA*-mut samples (enrichment score – 0.76) which was not significant (Appendix 1). Taken together we conclude that while *CLU* may play a pro-angiogenic role, we cannot make significant conclusions regarding the proangiogenic role of *CLU*+ CAFs in *BRCA*-mut PDAC.

Appendix 1. GSEA analysis of the “Hallmark Angiogenesis Pathway” in *BRCA*-mut vs. *BRCA*-WT stroma captured by LCM (positive values show enrichment in *BRCA*-mut samples). Normalized Enrichment score – 0.72, FDR q value – 0.86.

5) Is more nuclear HSF1 observed in murine sh*Brca2* tumors compared to control tumors? This would be a meaningful complement to patient data in Figure 3A-C and help support a causal role for cancer cell-intrinsic *BRCA* status in stromal activation of HSF1.

Response: We thank the reviewer for this suggestion, which we followed by staining KPC-shControl and KPC-sh*Brca2* murine tumors for CK (as a marker for cancer cells) and HSF1. Unfortunately, there is no good anti-mouse CLU antibody. We observed more nuclear HSF1 in the CK⁻ compartment of KPC-sh*Brca2* tumors compared to KPC-shControl (Appendix 2a). While this trend was not statistically significant, perhaps due to the very broad classification of CK⁺ vs CK⁻ compartments and due to the lack of an appropriate CLU antibody, it further supports a role for cancer cell-intrinsic *BRCA* status in stromal activation of HSF1.

Interestingly, we found a similar trend of HSF1 expression in the KPC-sh*Brca2* cancer cells (compared to KPC-shControl; Appendix 2B). In addition, mass-spec analysis of the secretome of KPC-sh*Brca2* cells compared to KPC-shControl revealed that 7 of the top 10 differentially secreted proteins are potential HSF1 targets (including CLU; Fig 4o discussed in our response to comment 3 above). It is beyond the scope of this manuscript to dissect the role of *BRCA* mutations in cell-autonomous regulation of HSF1 in the cancer cells. Nevertheless, these data as well as our new functional assays using CAFs isolated from KPC-shControl and KPC-sh*Brca2* tumors (detailed in our response to comment 1) strongly support a causal role for cancer-cell intrinsic *BRCA* status in stromal rewiring.

Appendix 2. HSF1 nuclear expression tends to be higher in *Brca*-deficient tumors. FFPE sections from KPC-sh*Brca2* or KPC-shControl tumors were stained by MxIF staining for HSF1, CK and DAPI. Cell segmentation was performed using Cellpose and cell classification by Qupath. The fraction of HSF1⁺CK⁻ out of all CK⁻ (a) and the fraction of HSF1⁺CK⁺ out of all CK⁺ (b) are presented.

6) Minor: In panel 6D, “CM” should be removed from the x-axis labels.

Response: Thank you for noting, this was removed.

Reviewer #2 (Remarks to the Author): with expertise in PDAC

In the manuscript by Shaashua et al., the authors investigate the effect of BRCA1/2 mutations in CAF populations. The authors compared CAFs in tumors from BRCA wild-type and mutant patients and show an increase in CLU⁺ positive CAFs compared with other CAF subtypes in BRCA-mutant patients. Based on the assumption that CLU is upregulated on the transcriptional level in BRCA-mutant tumors, the authors implicate the stress-responsive TF HSF1 in upregulation of CLU. The data to support transcriptional upregulation of CLU in BRCA-mutant tumors is thin, while the evidence for wholesale “transcriptional re-wiring” is nonexistent. Beyond that, the functional upshot of any change in CAFs that does occur in BRCA-mutant tumors is not explored.

Response: We thank the reviewer for the thorough assessment of our work. We addressed the reviewers’ concerns by performing new experiments and analyses that characterize the functional changes in CAFs of *BRCA*-mut tumors, link CLU to these changes, and strengthen the connection between *BRCA*-mut cancer cells and functional rewiring of CAFs.

Major Points

1. The authors suggest that BRCA-mutant tumors are enriched for CLU-positive CAFs, though their RNA-seq data in microdissected stromal samples seems to undermine this. First, CLU itself is not significantly upregulated in the stroma of BRCA-mutant tumors. Even if we acknowledge a trend towards higher CLU, the RNA-seq gives us little reason to believe that a wholesale shift in CAF subtypes is occurring in BRCA-mutant tumors. Only 40 genes are significantly altered between BRCA wild-type and mutant tumors. A cursory comparison of gene expression in CAF populations from Peng et al. (provided by the authors) with the altered genes in their RNA-seq seems to yield no alignment with a particular subtype of CAF. The authors should use GSEA to compare their RNA-seq with gene expression in the single cell data. Similarly, the altered genes in PSCs induced by Brca depletion in cancer cells bears almost no resemblance to stromal changes in BRCA-mutant tumors, again GSEA could show a correspondence. If neither comparison yields a similarity, we are left wondering about the relevance of the single cell and in vitro RNA-seq data the authors include.

Response: We thank the reviewer for this comment. While we see elevation in CLU⁺ CAFs in patient samples at the protein level, it was more challenging to validate it at the RNA level. This however is expected due to several reasons: First, our micro-dissected cohort is relatively small, and given patients’ variability has low statistical power. Ideally, we would have performed LCM on a larger cohort however this procedure was done on fresh-frozen samples which are extremely rare for the *BRCA*-mut population. Moreover, this procedure is technically challenging and laborious. Therefore, we RNA-sequenced only a small cohort of patients, but stained a much larger cohort. Second, due to post-translational modifications, changes at the protein levels are not always reflected as robustly at the RNA level. Finally, while CLU⁺ CAFs are significantly induced in the *BRCA*-mut tumors, this subtype is still less abundant than the SMA⁺ CAFs (as shown in Fig. 1b-c), which likely limits our ability to detect this change in the micro-dissected specimens which are comprised of bulk stroma. Indeed, GSEA analysis did not yield significant upregulation of the iCAF subpopulation from the Peng dataset in the LCM *BRCA*-mut samples.

We agree that the LCM method is limited, however, given the rarity of such a cohort of stromal *BRCA*-mut and *BRCA*-WT patient samples, we believe that this data is informative, nevertheless. With regards to the PSC data, it supports our finding of *Clu* as a major upregulated factor, as *Clu* was the 6th most DE gene in RNA-seq of PSCs following exposure to KPC-sh*Brca2*-CM compared to KPC-shControl-CM (Figure 5a and Supplementary Table 8).

To address the reviewers' comment we removed much of the PSC RNA-seq data from the text and from Figure 5 to Supplementary Figure 5, emphasizing the CAF data. We also modified the text to explain how differences between experimental systems and methodologies may lead to the observed differences in gene expression patterns (rows 506-507):

“PSCs are highly plastic and assume distinct transcriptional and functional properties depending on culture conditions (2D vs 3D, cancer CM etc).”

Finally, we highlight similarities that are now strongly supported by the new functional data added to the manuscript and detailed in our response to comment 3 below.

2. The authors provide IHC for MUCB and SERPINA1 and tumor-derived exosome data to “validate” their RNA-seq findings. This is irrelevant, as the question is not whether the proteins are expressed, but rather whether they are increased in *BRCA* mutant vs wild-type stroma. Indeed they are not, as stated in lines 341-342. The authors then resort to looking at MUC5B levels only in CLU+ CAFs, to finally find a difference. The RNA-seq data was obtained using whole stroma, so why should this validation require only looking at CLU+ CAFs, especially if CLU+ CAFs are so greatly increased in *BRCA*-mutant tumors?

Response: As commented above, while CLU⁺ CAFs are significantly induced in the *BRCA*-mut tumors, this subtype is still less abundant than the SMA⁺ CAFs (as shown in Fig. 1b-c), thus studying specifically the CLU⁺ CAFs increases the power of this analysis.

Furthermore, *SERPINA1* and *MUC5B* are secreted proteins and are difficult to detect via protein IHC; there is likely much expression than what we could assess with protein IHC, therefore we tried to detect *SERPINA1* and *MUC5B* at the RNA level by Fluorescence in situ hybridization (FISH). Unfortunately the staining of MUC5B technically failed. We were however able to detect *SERPINA1* puncta in FFPE samples of our patient cohort, and found a trend of higher expression of *SERPINA1* in *BRCA*-mut patients (Appendix 3). This result was not significant, perhaps due to the small sample size, and we therefore did not include it in the revised manuscript.

While we agree with the reviewer that this finding could be further validated in future studies on larger cohorts, this is not the focus of our current study. We do not overinterpret this data and/or claim that *SERPINA1* or *MUC5B* are *BRCA*-mut biomarkers. We also changed our wording from “validate” to “analyze” when referring to the IHC analysis of *SERPINA1* and *MUC5B* in the text (row 283). We mention this finding, which is a minor part of the manuscript, for future validation by us or others, due to the aforementioned rarity of our cohort. Moreover, this finding is independent from the main finding that *BRCA* mutations elicit an HSF1-dependent transcriptional program in PSCs to become CLU⁺ CAFs, and does not affect the main conclusions of our study.

Appendix 3. *SERPINA1* expression tends to be higher in the stroma of *BRCA*-mut patients. FFPE sections from 5 *BRCA*-mut and 4 *BRCA*-WT PDAC patients were stained using simultaneous hybridization chain reaction (HCR) immunohistochemistry (IHC) and HCR RNA fluorescence in situ hybridization (FISH) (Molecular Instruments). For HCR-RNA FISH, slides were labeled with a *SERPINA1* probe set and accompanying amplifiers. A negative control slide stained with antibodies and amplifiers, but not with RNA probes, served as the threshold to evaluate true HCR-FISH signal relative to background noise. Specific stromal regions of interest were selected to exclude cancer cells. The number of nuclei and the number of FISH puncta in each image were calculated using Fiji.

3. The authors fail to provide any evidence that the supposed changes in CAFs have any functional importance. The tumor growth changes shown in Figure 6D could as easily be due to tumor cell intrinsic reasons as to CAF-mediated changes.

Response: We performed several new experiments and analyses to address this important comment, including functional analyses of immune regulation and ECM modulation. Together, these new analyses support our hypothesis that *BRCA*-mut-associated CAFs are less myofibroblastic and more immune-regulatory.

- To assess immune-modulation, we isolated CD8⁺ T-cells from spleens of naïve C57BL/6 mice and activated them in the presence of CAFs isolated from WT or *BRCA2* deficient KPC tumors. We found that CAFs from KPC-sh*Brca2* tumors significantly repressed the activation CD8⁺ T-cells relative to CAFs from KPC-shControl tumors as measured by CD69 and CD25 surface expression (Figure 6b-c).
- In the original manuscript we have shown that PSCs upregulate PD-L1 in response to KPC-sh*Brca2* CM. We extended our search for immune-modulatory cell surface proteins by analyzing our RNA-seq data from mouse CAFs using the ICELLNET receptor-ligand analysis tool (Noel et al, 2021). We found that *Pvr* (CD155), the ligand for the inhibitory checkpoint receptor TIGIT, was upregulated in CAFs derived from KPC-sh*Brca2* tumors vs KPC-shControl tumors (Figure 6d). To test whether this leads to differential cell-surface expression we performed FACS staining for the TIGIT ligands CD155 and Nectin2, as well as PD-L1, on CAFs freshly derived from KPC-sh*Brca2* vs KPC-shControl tumors. CAFs from KPC-sh*Brca2* tumors exhibited significantly higher cell-surface expression levels of CD155 and Nectin2 (as compared to KPC-shControl), and a similar trend was observed for the expression of PD-L1 (Figure 6e-g). (Figure 6 e-g). Of note, Nectin2 (PVRL2) was also one of the upregulated immune regulatory genes in subcluster 2 (*CLU*^{igh}) of the human Peng PDAC data set, further supporting the link between *Clu* and immune regulatory functions in mouse and human CAFs.

These data suggest that BRCA-deficient cancer cells enhance an immune-regulatory phenotype of CAFs.

- Next, to further explore the changes in ECM deposition (that we have shown *in vitro* in the original manuscript) we assessed ECM organization in the vicinity of CLU⁺ or α SMA⁺ CAFs by performing second harmonic generation imaging in our patient cohort (Figure 6j-m). We found that the ECM associated with CLU⁺ CAFs is significantly different from that found in the vicinity of α SMA⁺ CAFs. Specifically, α SMA-associated ECM was stiff, while CLU-associated ECM was “curly”, as measured by decreased perpendicular alignment, and increased curvature and branching indices (Figure 6j-m).

Taken together, these new experiments provide vast evidence that the changes in CAFs have functional importance. This was added to rows 551-596 of the text and to a Figure 6 (see below).

In addition, to address the reviewers’ comment, we removed the tumor size figure panel (previously Figure 6d) as it may indeed reflect other processes which are independent of the changes we observe in the CLU⁺ CAFs.

Shaashua et al; Figure 6

Figure 6. *Brca2*-deficient cancer cells shift CAF functions. (a) Immortalized PSCs were seeded in Matrigel for 4 days in 10% FBS in DMEM. Conditioned media (CM) from PSCs or from KPC cells in which *Brca2* was silenced by shRNA or non-targeting shControl (KPC) was then added for an additional 4 days. 3nM of the HSF1 inhibitor, CMLD011866 (aglaroxin C), or PBS control was added to the conditioned media. *Pd11* were measured by qRT-PCR. Data are presented as mean \pm SEM. (b-c) CD8⁺ T-cells were isolated from spleens of naïve C57BL/6 mice, cocultured, and activated with anti-CD3/CD28 beads for 24 hours in the presence of CAFs isolated from either KPC-shControl or KPC-sh*Brca2* tumors. Subsequently, T-cells were subjected to FACS analysis for surface expression of CD69 and CD25. Data quantified are presented as mean \pm SEM of 6 KPC-shControl and 6 KPC-sh*Brca2* mice. (d) Scores of ligand-receptor binding were calculated using the ICELLNET R package (see Methods) to predict potential differential interactions between ligands of CAFs derived from shControl vs sh*Brca2* tumors with immune checkpoint receptors on CD8⁺ T-cells. (e-g) KPC-shControl and sh*Brca2* tumors were dissociated into single-cell suspensions. Dead cells were excluded using Draq7 staining. CD45⁺EpCAM⁺PDPN⁺ cells were isolated and stained with anti-CD155 (e), anti-Nectin2 (f), and anti-PD-L1 (g). Statistical significance was assessed by t-test. (h-i) PSCs were treated with CM derived from KPC-sh*Brca2* or KPC-shControl organoids, or with growth medium (DMEM with 5% FBS) as control for 4 days. Then, cultures were stained with Sirius red (SR) to assess collagen deposition. Collagen levels were calculated by measuring the absorbance at 540nm and 605nm wavelength (see Methods). Each point represents the average of sh*Brca2* or shControl normalized to the average of the growth medium control in each experiment. n=3 independent experiments, each representing an average of 5 technical replicates. t-test was performed on normalized values. (i) Representative images of PSCs stained with SR following 4 days treatment with CM derived from KPC-sh*Brca2* or KPC-shControl organoids. Scale bar=300 μ m. (j-m) FFPE tumor sections from *BRCA*-mut and *BRCA*-WT PDAC patients were stained by double staining for α SMA and CLU and imaged using Second harmonic generation (SHG) imaging. (j). Representative images are shown. DRAQ5 was used to stain nuclei. Scale bar- 100 μ m, or 25 μ m (inset). (k-m) Quantification of matrix pattern using the TWOMBLI plug-in (see Methods). The following parameters were analyzed: (k) Alignment - the extent to which fibers within the field of view are oriented in a similar direction; (l) Curvature- the mean change in angle moving incrementally along 40 μ m mask fibers; and (m) Branchpoint - the number of intersections of mask fibers in the image. (n) Schematic representation of the proposed model. Secreted factors from *BRCA*-mutated cancer cells induce *HSF1* activation in a subset of adjacent PSCs leading to their transcriptional rewiring into immune-regulatory CLU⁺ CAFs.

“The shift in CLU⁺/SMA⁺ CAF ratios in human patients, and the shift in Clu vs Acta2 expression in PSCs conditioned by Brca2-deficient cancer cells, suggest that CAFs of BRCA-mutated tumors may undergo a shift from myofibroblastic to immune-regulatory functions. Supporting this notion, Pd11, whose upregulation in PDAC CAFs was suggested to mediate T-cell immune suppression^{19, 84, 85}, was induced in PSCs by KPC-shBrca2 CM, but not by KPC-shControl CM (Figure 6a). Similar to Clu, the induction of Pd11 was inhibited by aglaroxin C, suggesting an increased immune-regulatory function for PSCs induced by BRCA-deficient cancer cells, in an HSF1-dependent manner (Figure 6a). To functionally test the ability of CAFs isolated from Brca2-deficient tumors to regulate immune cell function, we isolated CD8⁺ T-cells from spleens of naïve C57BL/6 mice, and activated them in the presence of CAFs from KPC-shControl or KPC-shBrca2 tumors (Figure 6b-c and Supplementary Figure 6a and b). T-cells activated in the presence of CAFs from Brca2-deficient tumors were significantly more repressed in their ability to upregulate the activation markers CD25 and CD69 compared to T-cells activated in the presence of CAFs from shControl tumors.

To search for potential factors that could be mediating the inhibitory effect of CAFs from shBrca2 tumors on T-cells, we mined our tumor-derived CAF RNA-seq data from KPC tumors using the ICELLNET receptor-ligand analysis tool employing a CD8⁺ T-cell-receptor dataset⁸⁶. ICELLNET is a computational tool that calculates a communication score for ligand-receptor interactions based on transcriptomic data and a database of potential ligand-receptor pairs. We applied ICELLNET to screen for immune modulatory surface ligands in CAFs that may inhibit T-cell activity, and found that CAFs from KPC-shBrca2 tumors scored higher than KPC-shControl CAFs

in the checkpoint signaling axis involving the TIGIT and CD96 T-cell inhibitory receptors, and their cognate ligands CD155 and Nectin1-3 (Figure 6d). To experimentally validate the cell-surface expression of inhibitory checkpoint markers on CAFs, we isolated CAFs from KPC-shControl or KPC-shBrca2 tumors, and performed FACS analysis using antibodies for PD-L1, CD155, and Nectin2. CAFs from KPC-shBrca2 tumors demonstrated higher cell surface expression of CD155 and Nectin2 (as compared to KPC-shControl), and a similar trend was observed for the expression of PD-L1 (Fig. 6e-g and Supplementary Fig. 6c), confirming the ICCELLNET receptor-ligand results and further supporting their role in suppressing T-cell function.

Next, we assessed myofibroblastic functions. To that end, we measured the ability of PSCs to secrete collagen, in-vitro, using Sirius Red staining. We found that PSCs conditioned by KPC-shBrca2 medium secreted significantly less collagen than PSCs conditioned by KPC-shControl medium (Figure 6h-i). To test the relevance of these findings in patients we assessed ECM organization in the vicinity of CLU⁺ or α SMA⁺ CAFs by second harmonic generation (SHG) imaging combined with MxIF staining in BRCA-WT vs BRCA-mut patients (Figure 6j-m). CLU-rich stromal regions that are abundant in BRCA-mut tumors demonstrated a noticeably altered ECM architecture, characterized by significantly reduced parallel alignment, and increased curvature and branching of collagen streaks compared to α SMA-rich regions (Figure 6j-m). Overall, our findings suggest that BRCA-mut cancer cells promote a stressful TME that leads to the activation of HSF1 in a subset of PSCs. These PSCs are reprogrammed into immune regulatory CLU⁺ CAFs, resulting in a different stromal landscape in BRCA-mut compared to BRCA-WT PDAC tumors (Figure 6n).”

Minor point

1. It is unclear whether the data shown in Figure 2F is from multiple patients or multiple fields of view. How many patients were compared?

Response: The data shown is from multiple patients. Each dot represents one patient, with a total of 9 patients per group, as detailed in the Figure legend (rows 314-318):

“(e-f) FFPE tumor sections from 9 BRCA-mut and 9 BRCA-WT PDAC patients were stained by MxIF using antibodies for the indicated proteins. DAPI was used to stain nuclei. Scale bar, 50 μ m. Representative images are shown in (e). MUC5B and SERPINA1 protein levels were quantified by ImageJ software and the area stained by each protein and CAF marker was measured. Quantification of MUC5B colocalization with CLU and α SMA, analyzed by two-way ANOVA, is presented as mean \pm SEM in (f).”

Reviewer #3 (Remarks to the Author): with expertise in CAF, bioinformatics

The manuscript by Shaashua, Pevsner-Fischer and colleagues describes differences in the stromal cell landscape of pancreatic cancer patients with and without BRCA1/2 mutations. Using H&E, IHC, MxIF and single-cell RNA-seq the authors identify several types of cancer-associated fibroblasts, of which a subset expressing Clusterin (CLU) is enriched in BRCA-mutated tumors.

Through a series of in vitro experiments, the authors provide evidence that this phenotype can be induced by transcription factor HSF1 in CAFs and leads to a switch from a myofibroblastic to an immune-regulatory phenotype of CAFs – mediated by BRCA-mutant tumor cell to fibroblast crosstalk.

Overall, this manuscript is a great example how the TME is at least in part shaped by tumor intrinsic features. It reveals that BRCA1/2 mutations (found in up to 8% of patients with sporadic pancreatic cancer and in much higher frequencies in breast and ovarian cancer) can directly modulate the TME, as shown through CAF phenotypes. As such, it represents a significant advancement for the field.

While I think the work is overall a strong mechanistic study, I would like to encourage the authors to take some of the points listed below into account before resubmitting the revised version of their article.

Response: We thank the reviewer for the helpful suggestions and positive assessment of our manuscript.

Major:

1. The authors analyze single-cell data from Peng et al. and find three major CAF subtypes. One of these is characterized by high levels of MHC II genes and is therefore considered antigen presenting (ap)CAF. Looking into Suppl. Table 2, I would interpret this cluster (cluster 3) as immune cell doublets. These cells express considerable amounts of CD45, CD3, CSF1R, CD14. Additionally, they also express MCAM (which is confusing because the main text says MCAM⁺ cells were excluded) and RGS5, so they might in fact be doublets of immune cells and pericytes. [Tools like scGate (PMID 35258562) to reliably “gate” single cell data may help to clean this up]. I am not convinced that the cluster the authors suggest as apCAF is not a technical artifact.

Response: We thank the reviewer for this comment, which we followed by performing an scGATE analysis of the data by Peng et al. This analysis showed that only 1.3% of the cells included in our analysis of the Peng dataset have immune-like features (Appendix Table. 1a). Indeed, these cells cluster in the MHC-II⁺ cluster, however they are still only a small fraction of this cluster (13%; Appendix Table. 1B) and their exclusion from the analysis did not change the UMAP or the significantly enriched genes in each cluster (Appendix 4). Importantly, the identity of the *CLU*⁺ clusters does not change with or without these cells. With this in mind, and given the increased appreciation of heterogeneity and plasticity in cell states, functions and origins in the TME, we believe that there is not sufficient justification to exclude these cells from the analysis.

Similarly, with regards to MCAM⁺ cells, we excluded from our analysis a distinct MCAM⁺ cluster that had differentially expressed levels of MCAM compared to the other clusters. We chose not to exclude additional MCAM⁺ cells (that can still be found mostly in the MHC-II⁺ cluster) since a recent study showed expression of MCAM in vascular fibroblasts (Zhang, 2020), suggesting that MCAM could be expressed in CAFs.

a

N = 6405	Non-Myeloid cells	Myeloid cells
Non-T cells	6318	25
T cells	62	0

b

N = 623	Non-Myeloid cells	Myeloid cells
Non-T cells	536	25
T cells	62	0

Appendix Table 1. scGATE analysis of immune cell characteristics in the dataset by Peng et al. Classification of "Pure" myeloid cells and T-cells was done utilizing pre-defined gating models of the scGate algorithm on the entire Peng dataset. (a) Count table of 6405 fibroblasts and stellate cells from the Peng et al. dataset. (b) Count table of the 623 cells of the MHC-II⁺ cluster (cluster 3) following model based scGate algorithm.

Appendix 4. Uniform Manifold Approximation and Projection (UMAP) of 6,318 cells from the Peng et al. dataset following exclusion of 87 cells that were classified as myeloid cells or T-cells by a pre-defined model of the scGate algorithm. Color-coded gene expression of MCAM and the indicated CAF markers is presented.

2. Given the point above, and the manuscript by some of the authors: Elyada et al (PMID 31197017) stating that ~20% of CAFs with detectable MHC II expression do not form a separate cluster, I am a little concerned about the distinct stains for HLA-DR, SMA and CLU presented in

figure 1 and Suppl. Figure 1. These results seem contradictory to the results in Elyada et al to me? For the stains, I am missing a positive control that marks all fibroblasts (eg PDGFRA) - the authors aim to exclude non fibroblast cells (CD45- Cytokeratin-). In the case of SMA for example we know that pericytes are a main source of SMA expression so this strategy has shortcomings (that may extend beyond this example).

Response: Thank you for this point. Following the reviewers' suggestion, we stained patient samples with antibodies against PDGFRA, however we did not observe universal labeling of all fibroblasts. This data was added to the manuscript in Supplementary Figure 1b, and rows 151-161 of the text. This result in line with Elyada et al, who demonstrated that the 3 CAF subtypes were comprised of both PDPN⁺PDGFR⁺ and PDPN⁺PDGFR⁻ cells (Figure 5c of Elyada et al, 2019).

Supplementary Figure 1. PDAC stroma is comprised of three distinct CAF subtypes. (b) FFPE tumor sections from 5 *BRCA*-WT and 3 *BRCA*-mut PDAC patients were stained by MxIF for PDPN, PDGFRA, S100A4, CLU, α SMA, CD45 and cytokeratin (CK). DAPI was used to stain nuclei.

“In particular, we stained for α -smooth-muscle-actin (α SMA), podoplanin (PDPN), platelet-derived growth factor receptor alpha (PDGFRA), human leukocyte antigen DR isotype (HLA-DR; an MHC class II molecule), and S100A4, all of which were previously described as CAF markers in different cancer types^{8, 39, 40, 54}. We also stained for clusterin (CLU), which was previously suggested as a marker of bone marrow derived fibroblasts in breast cancer⁴ (Figure 1a-b and Supplementary Figure 1a-b). Of these proteins, three marked discrete CAF subtypes (negative for CD45 and cytokeratin), and together covered most of the stromal cells – α SMA, CLU and HLA-DR (MHC-II; Figure 1a-b and Supplementary Figure 1c-e). S100A4 marked mostly CD45⁺ immune cells in this patient cohort, PDPN marked a subset of α SMA⁺ CAFs, and PDGFRA partially overlapped with other markers; therefore, these were not chosen for further analysis (Supplementary Figure 1a-b).”

Regarding the apCAFs, while these are not the focus of our study, and their clustering does not affect our conclusions about the CLU⁺ immune-regulatory CAFs, we are happy to clarify this issue. Different studies have suggested different clustering of the apCAFs as either a separate cluster or as part of other CAF clusters. In Elyada et al. there were differences between the human

and mouse datasets – while apCAFs formed a separate cluster in the mouse, this was not observed in human samples. Moreover, while *Clu* was an apCAF marker in the KPC mouse model, it was the 5th ranked iCAF marker in the human dataset (Supplementary Table S13 in Elyada et al; and is now also mentioned in our manuscript rows 163-166; see below). These discrepancies could have resulted from the relatively small number of patients used in the study from Elyada compared to the Peng study. Recently, Huang et al (Cancer Cell 2022; PMID: 35523176) demonstrated that apCAFs in PDAC originate from mesothelial cells which upregulate fibroblastic features. Integrating their mouse apCAF signature with human scRNA-seq data sets, they found that human apCAFs formed a separate cluster from iCAFs and myCAFs. These data in addition to our negative staining of CD45 and cytokeratin, and our scRNA seq clustering results, support the existence of these distinct clusters in PDAC.

“Clu was shown to be expressed by α SMA^{low} CAFs in breast and pancreatic cancer^{8, 13, 40}, however the identity of these α SMA^{low} CAFs was not fully elucidated – in mouse models of PDAC Clu was shown to be expressed by apCAFs, whereas in human patient samples it is expressed by inflammatory CAFs¹².”

3. I am concerned about the conclusion the authors draw from Fig 4H that “Treatment with aglaroxin C abolished the induction of Muc5b expression”. The comparison between shBrca2 + vs - aglaroxin is not significant and 2 of the 5 replicates in the + group show a comparable or higher Muc5b level to what is observed as the median in the – group. That is actually along the lines of their other experiment: “Additionally, Muc5B showed somewhat reduced expression in Hsf1 null mice but this result was not significant”. Did the authors also perform a targeted immunoprecipitation PCR experiment for Muc5b promoter binding by HSF1 similar to what is shown for *Clu* (Fig. 4L)?

Response: To address this comment we conducted a targeted ChIP-PCR experiment in which we assessed *Muc5B*-promoter binding to HSF1. We found that the *Muc5B* is not significantly bound to HSF1 (Appendix 5). This suggests that the increase in *MUC5B* in *BRCA*-mut stroma is indirectly associated with HSF1. This result is in line with the HSF1-targets-database analysis (HSF1base.org) which did not indicate a direct binding of HSF1 to the *Muc5b* gene (as opposed to *Clu*).

Appendix 5. HSF1 does not significantly bind to the promoter of *Muc5B*. Immortalized PSCs were cultured with or without KPC-shControl-CM for 24 hours. ChIP-PCR was performed for putative heat-shock elements of *Muc5b* (and *Hsp70*, as positive control) following pulldown with anti-HSF1 antibody compared to IgG control. Binding data was normalized to the binding to an intergenic promoter, and is presented as mean \pm SEM.

4. I am wondering if the authors can provide any insights into what drives differential HSF1 activation in fibroblasts between BRCA1/2 WT and mutant tumors? Do the differentially expressed genes in tumor cells from their mouse model (KPC-shBrca2 and KPC469 shControl cells) give any insights what might be secreted by one but not the other? Have the authors considered to mass spec the conditioned media?

Response: We thank the reviewer for this suggestion, which we followed by performing the suggested mass spec analysis. To assess what cancer-secreted factors may contribute to activation of HSF1 and *Clu* in CAFs, we prepared CM from KPC-shCont and KPC-shBrca2 organoids. To assess whether the effect on *Clu* expression is mediated by a protein, we boiled the CM, and found that this heat inactivation significantly reduced upregulation of *Clu* (Figure 4n). Next, we performed mass spec analysis of the CM. This analysis revealed that the two most differentially secreted proteins from KPC-shBRCA2 vs KPC-shCont organoids were the Regenerating islet-derived (Reg) proteins, REG3G/B (the top ten differentially expressed in BRCA CM compared to NT CM are shown below, the full list is in Supp Table 7). Reg proteins play key roles in pancreatic inflammation and carcinogenesis through different pathways, including STAT3 and RAF-MEK-ERK signaling and immune suppression (Liu et al., 2017, Li et al. 2016, Gironella et al., 2013, Li et al., 1996, Zhang et al., 2021). Moreover, we have previously demonstrated an association between REG3B/G and HSF1 signaling, by showing that REG3B/G are upregulated during inflammation in the colon in an HSF1-dependent manner (Levi-Galibov et al., 2020). In fact, six of the ten most differentially secreted proteins from KPC-shBrca2 vs KPC-shCont organoids were previously shown by us to be upregulated during colon inflammation in an HSF1-dependent manner (REG3G, REG3B, GC, SERPINH1, FN1, and PXDN; Levi-Galibov et al., 2020). We also found CLU itself in this list. HSF1 is well known to be activated in cancer cells. Supporting this notion, we see a trend of higher HSF1 expression in KPC-shBrca2 vs KPC-shCont cancer cells in tumors (Appendix 6). It is beyond the scope of this study to investigate the cell-intrinsic role of loss-of-*Brca2* in regulation of HSF1 in cancer cells. Nevertheless, these findings suggest that loss of BRCA2 leads to differential secretion of proteins that lead to activation of HSF1 in stromal fibroblasts.

This data was added to the manuscript in Figure 4n-o, and to the text (rows 422-440) as follows:

“In search for factors that may mediate this effect, we next analyzed the medium conditioned by KPC-shBrca2 cells. First, to test whether HSF1 and Clu upregulation is mediated by secretion of proteins, we boiled CM from KPC-shControl and KPC-shBrca2 organoids and treated PSCs with either un-boiled or boiled CM. Boiling of KPC-shBrca2 CM significantly reduced expression of Clu by PSCs (Fig. 4n), suggesting that this effect is mediated by a secreted protein(s). Next, we performed mass-spectrometry analysis of the organoid CM. The two most differentially secreted proteins from KPC-shBrca2 relative to KPC-shControl organoids were the Regenerating islet-derived (Reg) proteins, REG3B/G (Fig. 4o, Supplementary Table 7). REG3B/G are C-type secreted lectins that play active roles in pancreatitis and in the transition from pancreatitis to

pancreatic cancer through different mechanisms, including induction of STAT3, RAF-MEK-ERK signaling, and immune cell modulation^{81, 82, 83, 84}. Of note, we have previously demonstrated an association between REG3B/G and HSF1 signaling, by showing that REG3B/G are upregulated during inflammation in the colon in an HSF1-dependent manner⁵⁴. In fact, six of the ten most differentially secreted proteins from KPC-shBrca2 vs KPC-shCont organoids were previously shown by us to be upregulated during colon inflammation in an HSF1-dependent manner (REG3G, REG3B, GC, SERPINH1, FN1, and PXDN)⁵⁴. We also found CLU itself in this list. These findings suggest that loss of BRCA2 in cancer cells leads to differential secretion of proteins resulting activation of HSF1 in stromal fibroblasts, and, potentially, also in the cancer cells themselves.”

Figure 4. (n) PSCs were cultured in the presence of boiled or unboiled CM from KPC-shControl and KPC-shBrca2 organoids as described in (g-j). Expression levels of *Clu* in PSCs were subsequently measured by qRT-PCR. (o) Top 10 differentially expressed proteins (fold change shBrca2/shControl) from CM of KPC-shControl and KPC-shBrca2 organoids as measured by mass-spectrometry.

Appendix 6. HSF1 nuclear expression tends to be higher in Brca-deficient tumors. 20,000 shBrca2- or shControl-KPC cells were injected into the pancreas of 8-week-old mice. Tumors were harvested 2-3 weeks following cells injection and FFPE embedded. 5µm sections were then stained by MxIF staining for HSF1, CK and DAPI. Cell segmentation was performed using Cellpose and cell classification by Qupath. The fraction of HSF1⁺CK⁺ out of all CK⁺ is presented.

Minor:

1. The authors excluded the normal adjacent tissue samples from their single-cell analysis. How is *CLU* expressed in normal adjacent tissue? Are the *CLU* high and low clusters from the same patients? Or are there between patient differences (e.g. some only have high, others only have *CLU* low CAFs)?

Response: To address these interesting points we checked the expression levels of *CLU* in the normal control samples as defined by Peng et al. In this study there were no normal adjacent samples; the normal controls were from patients with non-pancreatic malignancies or from non-malignant pancreatic cysts. We analyzed all cells that were defined as either stellate cells or fibroblasts by Peng et al. in control tissue (Appendix 7; left panel) and compared this analysis to our analysis of *CLU* expression in tumor tissue (Appendix 7, right panel). We found that *CLU* was indeed expressed in control tissue, however its expression was scattered in all clusters, as opposed to the tumor tissue where *CLU* was expressed by a specific cluster of cells.

To examine the intra patient dispersion of *CLU*^{high} vs. *CLU*^{low} fibroblasts, we color-coded the UMAP based on patient numbers. We found that the different clusters of *CLU* do not stem from differences between patients.

This new data was added to the manuscript (Supplementary Fig. 1n and rows 210-211 of the text):

“This segregation was evident across patients, and did not stem from intra-patient variability (Supplementary Fig. 1n).”

Appendix 7. *CLU* is expressed in specific clusters in CAFs, whereas its expression is scattered across clusters in fibroblasts from non-malignant tissue. Fibroblast and stellate cells defined by the Peng et al dataset are presented by Uniform Manifold Approximation and Projection (UMAP). Expression of *CLU* in 1458 cells from control sample (left panel), and 6,405 cells from tumor samples (right panel) is shown.

Supplementary Figure 1n. UMAP dot plot of 6,405 cells defined as fibroblast and stellate cells in the dataset from Peng et al, color-coded by patient number.

2. I don't understand how the top 20 genes for figure 5E were chosen. When I sort suppl Table 9 tab 1 by adjusted p-value, I see a different set of genes. Can the authors also comment on the purity of their sort (CSF1R and KRT7/8/18 come up as quite highly and differentially expressed and upregulated in shBRCA2). Maybe the authors could plot the expression of a few general fibroblasts markers (Pdgfra, Pdpn, ..) to show they are similar between the groups.

Response: We thank the reviewer for this comment. As stated on the top of tab 4, the top-20 list was calculated after filtering for genes with BaseMean larger than the 1st quartile ("This list represent the top DE genes after filtering for genes with BaseMean larger than the 1st quartile."), while the list of genes in tab 1 was filtered for genes with BaseMean >5. Also, the genes were sorted based on Log2 fold change and not by the p-value. We chose to apply this filtering method to exclude the many genes with low expression levels that were less likely to indicate significant biological differences.

We have now edited the comment on Tab 4 to clarify this as follows: *"This list represent the top DE genes based on log2 fold-change values, after filtering for genes with BaseMean larger than the 1st quartile."*

With regards to the purity of the CAF sorting, we utilized a stringent gating strategy to ensure that only PDPN⁺ CAFs (in mice PDPN serves as a good universal marker for CAFs) are isolated for sequencing (Supplementary Figure 5g). As detailed in our response to Major comment 1 above, we believe that the expression of markers such as *Csf1R* and even *Krt* does not suggest contamination but rather intermediate cell states. Supporting this notion, recent studies have shown that *CSF1R* is upregulated by CAFs and enhances their tumor promoting phenotype (Kosti et al., 2022, Kumar et al., 2017). Other studies have shown that *Krt8* and *Krt18* (Elyada et al., 2019, Huang et al., 2022) are markers of mesothelial-derived apCAFs.

Nevertheless, to address the reviewers' comment we plotted the expression of a few general fibroblast markers and found that their expression levels are similar between CAFs from KPC-shControl and KPC-sh*Brca2* tumors (Appendix 8). We also compared the expression of *Krt8* and *Krt18* in CAFs vs KPCs and found that while the expression of these genes is higher in CAFs from KPC-shControl compared to CAFs from KPC-sh*Brca2* tumors, the relative expression of these genes in either type of CAFs is 9-fold lower for *Krt8* and 20-fold lower for *Krt18* than in KPC cells (Appendix 9).

Appendix 8. The expression of general CAF markers is similar between CAFs from KPC-shControl and KPC-sh*Brca2* tumors. Expression of the indicated CAF markers in CAFs from sh*Brca2* and shControl tumors was assessed in CAF RNA-seq expression (Fig. 5 c-d) values and normalized to the average expression of the housekeeping genes *Actb*, *Hprt* and *Gapdh*. The results are log transformed.

Appendix 9. The relative expression of *Krt8* and *Krt18* is significantly lower in CAFs isolated from KPC-sh*Brca2* and KPC-shControl tumors compared to KPC cancer cells. The expression of *Krt8* and *Krt18* in CAFs from KPC-sh*Brca2* and KPC-shControl tumors and in the KPC cells themselves was assessed in CAF RNA-seq expression levels (Fig. 5 c-d) and normalized to the average expression of the housekeeping genes *Actb*, *Hprt* and *Gapdh*. The results are log transformed.

3. “We found that the expression of *Clu* was significantly lower in *Hsf1* null PSCs compared to WT PSCs, while the expression of other CAF markers, such as *Acta2* and *Il6*, was not altered (Figure 4E-F).” – How does this relate to expression correlation in their human laser capture microdissected CAF-rich region RNA-seq data? Is there a correlation between HSF1 and CLU expression levels?

Response: To address this comment we tested the correlation between *CLU* and *HSF1* levels in our LCM-RNA seq data which resulted in a non-significant positive correlation (Appendix 10). We believe that this result (p value= 0.13) is non-significant due to the small sample size and since the regulation of HSF1’s activity is mostly post-translational, through nuclear translocation. Thus, its transcript levels might fail to represent its activation in these samples.

Appendix 10. Pearson correlation between *CLU* expression and *HSF1* expression of LCM-captured stroma.

4. Fig. 6D: Maybe I misunderstood, but I assume cells were orthotopically implanted into C57BL/6J mice. In that scenario the x axis label should not refer to CM?

Response: We thank the reviewer for noticing this typo, which is now corrected in Figure 6.

Reviewers' Comments:

Reviewer #1:

Remarks to the Author:

The authors have thoroughly and meaningfully addressed my comments from the first submission.

Reviewer #2:

Remarks to the Author:

The authors have sufficiently addressed my concerns.

Reviewer #3:

Remarks to the Author:

I want to thank the authors for addressing most of my comments and find the manuscript significantly improved. I would like to still emphasize that my point/comment on the data presented in figure 4H has not been addressed. The statement in the results section that "Treatment with aglaroxin C abolished the induction of Muc5b expression" is misleading. The relevant comparisons for the point the authors are trying to make are not significant. There is no significant difference between KPC-shBrca2 and KPC-shBrca2 with aglaroxin C. The same holds true for KPC-shCont vs KPC-shCont with aglaroxin C. Therefore, aglaroxin C did not impact Muc5b expression in either of the two relevant comparisons. That makes the statement ".. [suggesting] that HSF1 regulates the expression of these genes" incorrect for the case of Muc5b. The only conclusion from 4H is that shBrca2 leads to increased Muc5b expression. A HSF1-dependency cannot be concluded from this data (or the additional data presented by the authors as part of their revisions in Appendix 5).

I strongly encourage the authors to remove the statement regarding HSF1 dependent regulation of Muc5b.

Response to reviewer comments

Reviewer #3

I want to thank the authors for addressing most of my comments and find the manuscript significantly improved. I would like to still emphasize that my point/comment on the data presented in figure 4H has not been addressed. The statement in the results section that “Treatment with aglaroxin C abolished the induction of Muc5b expression” is misleading. The relevant comparisons for the point the authors are trying to make are not significant. There is no significant difference between KPC-shBrca2 and KPC-shBrca2 with aglaroxin C. The same holds true for KPC-shCont vs KPC-shCont with aglaroxin C. Therefore, aglaroxin C did not impact Muc5b expression in either of the two relevant comparisons. That makes the statement “.. [suggesting] that HSF1 regulates the expression of these genes” incorrect for the case of Muc5b. The only conclusion from 4H is that shBrca2 leads to increased Muc5b expression. A HSF1-dependency cannot be concluded from this data (or the additional data presented by the authors as part of their revisions in Appendix 5). I strongly encourage the authors to remove the statement regarding HSF1 dependent regulation of Muc5b.

Response: We thank the reviewer for the helpful comments and for the positive assessment of the manuscript.

To address the reviewer’s last comment we rephrased the sentence concerning Muc5b in Figure 4. The revised sentence reads as follows (row 408):

“Treatment with aglaroxin C abolished the induction of Clu expression, suggesting that this induction is HSF1-dependent, and that HSF1 regulates the expression of these genes.”